# Fair Graph Message Passing with Transparency

## Abstract

Recent advanced works achieve fair representations and predictions through regularization, adversarial debiasing, and contrastive learning in graph neural networks (GNNs). These methods *implicitly* encode the sensitive attribute information in the well-trained model weight via *backward propagation*. In practice, we not only pursue a fair machine learning model but also lend such fairness perception to the public. For current fairness methods, how the sensitive attribute information usage makes the model achieve fair prediction still remains a black box. In this work, we first propose the concept *transparency* to describe *whether* the model embraces the ability of lending fairness perception to the public *or not*. Motivated by the fact that current fairness models lack of transparency, we aim to pursue a fair machine learning model with transparency via *explicitly* rendering sensitive attribute usage for fair prediction in *forward propagation* . Specifically, we develop an effective and transparent Fair Message Passing (FMP) scheme adopting sensitive attribute information in forward propagation. In this way, FMP explicitly uncovers how sensitive attributes influence final prediction. Additionally, FMP scheme can aggregate useful information from neighbors and mitigate bias in a unified framework to simultaneously achieve graph smoothness and fairness objectives. An acceleration approach is also adopted to improve the efficiency of FMP. Experiments on node classification tasks demonstrate that the proposed FMP outperforms the state-of-the-art baselines in terms of fairness and accuracy on three real-world datasets. The code is available in https://anonymous.4open.science/r/FMP-AD84.

## 1 Introduction

Graph neural networks (GNNs) (Kipf & Welling, 2017; Veličković et al., 2018; Wu et al., 2019; Han et al., 2022a;b) are widely adopted in various domains, such as social media mining (Hamilton et al., 2017), knowledge graph (Hamaguchi et al., 2017) and recommender system (Ying et al., 2018), due to remarkable performance in learning representations. Graph learning, a topic with growing popularity, aims to learn node representation containing both topological and attribute information in a given graph. Despite the outstanding performance in various tasks, GNNs often inherit or even amplify societal bias from input graph data (Dai & Wang, 2021). The biased node representation largely limits the application of GNNs in many high-stake tasks, such as job hunting (Mehrabi et al., 2021) and crime ratio prediction (Suresh & Guttag, 2019). Hence, bias mitigation that facilitates the research on fair GNNs is in an urgent need.

Many existing works achieving fair prediction in graph either rely on regularization (Jiang et al., 2022), adversarial debiasing (Dai & Wang, 2021), or contrastive learning (Zhu et al., 2020; 2021b; Agarwal et al., 2021; Kose & Shen, 2022). These methods adopt sensitive attribute information in training loss refinement. In this way, such sensitive attribute can be implicitly encoded in well-trained model weight through backward propagation. However, only achieving fair model is insufficient in practice since the fairness should also *lend perception to the public* (e.g., the auditors, or the maintainers of machine learning systems). In other words, the influence of sensitive attributes should be easily probed for public. We name such property for public probing as *transparency*. Specifically, we provide the following formal statement on *transparency* in fairness:

*Transparency in fairness: Onlookers can verify the released fair model with*

• *Transparent influence: How and if the sensitive attribute information influence fair model prediction.*

- *Less is more: The required resources to obtain the influence of sensitive attributes only includes well-trained model and test data samples* [1].

From auditors' perspective, even though the fairness metric for machine learning model is low, such fair model is still untrustful if the auditors cannot understand how the sensitive attributes are adopted to achieve fair prediction given the well-trained model. From maintainer's perspective, it is important to understand how the model provide fair prediction. Such understanding could help maintainers improve models and further convince auditors in terms of fairness. In summary, transparency aims to make fairness implementation understandable. The transparency aims to make the process of achieving fair model via sensitive attribute informations white-box [2]. Therefore, the maintainers and auditors both get benefits from model transparency. More importantly, similar to intrinsic explainability of the model [3], the fairness with transparency property is binary, i.e., the prediction model either embrace the fairness with transparency or not.

Based on the formal statement on transparency in fairness, the key rule to determinate whether a fair model is transparent is that the model prediction difference under the cases with and without sensitive attribute information can be identified given the well-trained fair model and test data samples. Unfortunately, many existing fairness works, including regularization, adversarial debiasing, and contrastive learning, do not satisfy transparency requirements in practice. For example, the fair model trained based on existing works are not with transparent influence. This is because the sensitive attribute is implicitly encoded in well-trained model weight. Therefore, it is intractable to infer how sensitive attribute influence the well-trained model weight without access the dynamic model training process. Additionally, for fair model obtained from existing works, the required resources for transparent influence includes training data and training strategy so that the influence of sensitive attributed can be probed via detecting well-trained model weight difference. In a nutshell, the current fair models based on loss refinement *lack of transparency*. A natural question is raised:

*Can we find fair prediction model with transparency?*

In this work, we provide a positive answer via chasing transparency and fairness in message passing of GNNs. The key idea of achieving transparency is to explicitly adopt sensitive attribute in message passing (forward propagation). Specifically, we design an fair and transparent message passing scheme for GNNs, called fair message passing (FMP). First, we formulate an optimization problem that integrates fairness and prediction performance objectives. Then, we solve the formulated problem via Fenchel conjugate and gradient descent to generate fair-and-predictive representation. We also interpret the gradient descent as aggregation first and them debiasing. Finally, we integrate FMP in graph neural networks to achieve fair and accurate prediction for node graph classification task. Further, we demonstrate the superiority of FMP by examining its effectiveness and efficiency, where we adopt the property of softmax function to accelerate the gradient calculation over primal variables.

In short, the contributions can be summarized as follows:

- We consider fairness problem from a new perspective, named transparency, i.e., the sensitive attribute should be easily probed for public. We point out that many existing fairness method cannot achieve transparency.

- We propose FMP to achieve fairness with transparency via using sensitive attribute information in message passing. Specifically, we use gradient descent to chasing graph smoothness and fairness in a unified optimization framework. An acceleration method is proposed to reduce gradient computational complexity with theoretical and empirical validation.

- The effectiveness and efficiency of FMP are experimentally evaluated on three real-world datasets. The results show that compared to the state-of-the-art, our FMP exhibits a superior trade-off between prediction performance and fairness with negligibly computation overhead.

---

[1] A naive way for many existing works (e.g., adding fair regularization, adversarial debiasing, et.al.) to obtain the influence of sensitive attribute is to train fair and unfair model with/without the sensitive attribute information, and then get the prediction difference. Therefore, the required resources includes training data and additional (unfair) model training.

[2] Similar with the goal of model explainability, only achieving accurate prediction is insufficient, and chasing explainability can help experts understand how the model provide prediction and convince users.

[3] Decision tree is intrinsic explainable while deep neural networks is not.

## 2 PRELIMINARIES

### 2.1 NOTATIONS

We adopt bold upper-case letters to denote matrix such as $\mathbf{X}$, bold lower-case letters such as $\mathbf{x}$ to denote vectors, calligraphic font such as $\mathcal{X}$ to denote set. Given a matrix $\mathbf{X} \in \mathbb{R}^{n \times d}$, the $i$-th row and $j$-th column are denoted as $\mathbf{X}_i$ and $\mathbf{X}_{\cdot,j}$, and the element in $i$-th row and $j$-th column is $\mathbf{X}_{i,j}$. We use the Frobenius norm, $l_1$ norm of matrix $\mathbf{X}$ as $||\mathbf{X}||_F = \sqrt{\sum_{i,j} \mathbf{X}_{i,j}^2}$ and $||\mathbf{X}||_1 = \sum_{ij} |\mathbf{X}_{ij}|$, respectively. Given two matrices $\mathbf{X}, \mathbf{Y} \in \mathbb{R}^{n \times d}$, the inner product is defined as $\langle \mathbf{X}, \mathbf{Y} \rangle = tr(\mathbf{X}^\top \mathbf{Y})$, where $tr(\cdot)$ is the trace of a square matrix. $SF(\mathbf{X})$ represents softmax function with a default normalized column dimension. Let $\mathcal{G} = \{\mathcal{V}, \mathcal{E}\}$ be a graph with the node set $\mathcal{V} = \{v_1, \cdots, v_n\}$ and the undirected edge set $\mathcal{E} = \{e_1, \cdots, e_m\}$, where $n, m$ represent the number of node and edge, respectively. The graph structure $\mathcal{G}$ can be represented as an adjacent matrix $\mathbf{A} \in \mathbb{R}^{n \times n}$, where $\mathbf{A}_{ij} = 1$ if existing edge between node $v_i$ and node $v_j$. $\mathcal{N}(i)$ denotes the neighbors of node $v_i$ and $\tilde{\mathcal{N}}(i) = \mathcal{N}(i) \cup \{v_i\}$ denotes the self-inclusive neighbors. Suppose that each node is associated with a $d$-dimensional feature vector and a (binary) sensitive attribute, the feature for all nodes and sensitive attribute are denoted as $\mathbf{X}_{ori} = \mathbb{R}^{n \times d}$ and $\mathbf{s} \in \{-1, 1\}^n$. Define the sensitive attribute incident vector as $\Delta_{\mathbf{s}} = \frac{\mathbb{1}_{>0}(\mathbf{s})}{||\mathbb{1}_{>0}(\mathbf{s})||_1} - \frac{\mathbb{1}_{>0}(-\mathbf{s})}{||\mathbb{1}_{>0}(-\mathbf{s})||_1}$ to normalize each sensitive attribute group, where $\mathbb{1}_{>0}(\mathbf{s})$ is an element-wise indicator function.

### 2.2 GNNs AS GRAPH SIGNAL DENOISING

A GNN model is usually composed of several stacking GNN layers. Given a graph $\mathcal{G}$ with $N$ nodes, a GNN layer typically contains feature transformation $\mathbf{X}_{trans} = f_{trans}(\mathbf{X}_{ori})$ and aggregation $\mathbf{X}_{agg} = f_{agg}(\mathbf{X}_{trans}|\mathcal{G})$, where $\mathbf{X}_{ori} \in \mathbb{R}^{n \times d_{in}}$, $\mathbf{X}_{trans}, \mathbf{X}_{agg} \in \mathbb{R}^{n \times d_{out}}$ represent the input and output features. The feature transformation operation transforms the node feature dimension, and *feature aggregation*, updates node features based on neighbors' features and graph topology. Recent works (Ma et al., 2021b; Zhu et al., 2021a) have established the connections between many feature aggregation operations in representative GNNs and a graph signal denoising problem with Laplacian regularization. Here, we only introduce GCN/SGC as an examples to show the connection from a perspective of graph signal denoising. The more discussions are elaborated in Appendix G. Feature aggregation in Graph Convolutional Network (GCN) or Simplifying Graph Convolutional Network (SGC) is given by $\mathbf{X}_{agg} = \tilde{\mathbf{A}} \mathbf{X}_{trans}$, where $\tilde{\mathbf{A}} = \tilde{\mathbf{D}}^{-\frac{1}{2}} \hat{\mathbf{A}} \tilde{\mathbf{D}}^{-\frac{1}{2}}$ is a normalized self-loop adjacency matrix $\hat{\mathbf{A}} = \mathbf{A} + \mathbf{I}$, and $\tilde{\mathbf{D}}$ is degree matrix of $\tilde{\mathbf{A}}$. Recent works (Ma et al., 2021b; Zhu et al., 2021a) provably demonstrate that such feature aggregation is equivalent to one-step gradient descent to minimize $tr(\mathbf{F}^\top (\mathbf{I} - \tilde{\mathbf{A}})\mathbf{F})$ with initialization $\mathbf{F} = \mathbf{X}_{trans}$.

## 3 FAIR MESSAGE PASSING

In this section, we propose a new message passing scheme to aggregate useful information from neighbors while debiasing representation bias. Specifically, we formulate fair message passing as an optimization problem to pursue *smoothness* and *fair* node representation simultaneously. Together with an effective and efficient optimization algorithm, we derive the closed-form fair message passing. Finally, the proposed FMP is shown to be integrated in fair GNNs at three stages, including transformation, aggregation, and debiasing step, as shown in Figure 1.

### 3.1 THE OPTIMIZATION FRAMEWORK

To achieve graph smoothness prior and fairness in the same process, a reasonable message passing should be a good solution for the following optimization problem:

$$\min_{\mathbf{F}} \underbrace{\frac{\lambda_s}{2} tr(\mathbf{F}^T \tilde{\mathbf{L}} \mathbf{F}) + \frac{1}{2}||\mathbf{F} - \mathbf{X}_{trans}||_F^2}_{h_s(\mathbf{F})} + \underbrace{\lambda_f ||\Delta_s SF(\mathbf{F})||_1}_{h_f(\Delta_s SF(\mathbf{F}))}. \tag{1}$$

where $\tilde{\mathbf{L}}$ represents normalized Laplacian matrix, $h_s(\cdot)$ and $h_f(\cdot)$ denotes the smoothness and fairness objectives [4], respectively, and $\mathbf{X}_{trans} \in \mathbf{R}^{n \times d_{out}}$ is the transformed $d_{out}$-dimensional node features and $\mathbf{F} \in \mathbf{R}^{n \times d_{out}}$ is the aggregated node features of the same matrix size. The first two terms preserve the similarity of connected node representation and thus enforces graph smoothness. The last term enforces fair node representation so that the average predicted probability between groups of different sensitive attributes can remain constant. The regularization coefficients $\lambda_s$ and $\lambda_f$ adaptively control the trade-off between graph smoothness and fairness.

**Smoothness objective $h_s(\cdot)$.** The adjacent matrix in existing graph message passing schemes is normalized for improving numerical stability and achieving superior performance. Similarly, the graph smoothness term requires normalized Laplacian matrix, i.e., $\tilde{\mathbf{L}} = \mathbf{I} - \tilde{\mathbf{A}}$, $\tilde{\mathbf{A}} = \hat{\mathbf{D}}^{-\frac{1}{2}} \hat{\mathbf{A}} \hat{\mathbf{D}}^{-\frac{1}{2}}$, and $\hat{\mathbf{A}} = \mathbf{A} + \mathbf{I}$. From edge-centric view, smoothness objective enforces connected node representation to be similar since $tr(\mathbf{F}^T \tilde{\mathbf{L}} \mathbf{F}) = \sum_{(v_i, v_j) \in \mathcal{E}} || \frac{\mathbf{F}_i}{\sqrt{d_i + 1}} - \frac{\mathbf{F}_j}{\sqrt{d_j + 1}} ||_F^2$, where $d_i = \sum_k A_{ik}$ represents the degree of node $v_i$.

**Fairness objective $h_f(\cdot)$.** The fairness objective measures the bias for node representation after aggregation. Recall sensitive attribute incident vector $\Delta_{\mathbf{s}}$ indicates the sensitive attribute group and group size via the sign and absolute value summation. Recall that the sensitive attribute incident vector as $\Delta_{\mathbf{s}} = \frac{\mathbb{1}_{>0}(\mathbf{s})}{||\mathbb{1}_{>0}(\mathbf{s})||_1} - \frac{\mathbb{1}_{>0}(-\mathbf{s})}{||\mathbb{1}_{>0}(-\mathbf{s})||_1}$ and $SF(\mathbf{F})$ represents the predicted probability for node classification task, where $SF(\mathbf{F})_{ij} = \hat{P}(y_i = j | \mathbf{X})$. Furthermore, we can show that our fairness objective is actually equivalent to demographic parity, i.e., $\left( \Delta_s SF(\mathbf{F}) \right)_j = \hat{P}(y_i = j | \mathbf{s}_i = 1, \mathbf{X}) - \hat{P}(y_i = j | \mathbf{s}_i = -1, \mathbf{X})$. Please see proof in Appendix B. In other words, our fairness objective, $l_1$ norm of $\Delta_s SF(\mathbf{F})$ characterizes the predicted probability difference between two groups with different sensitive attribute. Therefore, our proposed optimization framework can pursue graph smoothness and fairness simultaneously.

## 3.2 Algorithm for Fair Message Passing

For smoothness objective, many existing popular message passing scheme can be derived based on gradient descent with appropriate step size choice (Ma et al., 2021b; Zhu et al., 2021a). However, directly computing the gradient of the fairness term makes the closed-form gradient complicated since the gradient of $l_1$ norm involves the sign of elements in the vector.

### 3.2.1 Bi-level Optimization Problem Formulation.

To solve this optimization problem in a more effective and efficient manner, Fenchel conjugate (Rockafellar, 2015) is introduced to transform the original problem as bi-level optimization problem. Fenchel conjugate (Rockafellar, 2015) (a.k.a. convex conjugate) is the key tool to transform the original problem into bi-level optimization problem. For the general convex function $h(\cdot)$, its conjugate function is defined as $h^*(\mathbf{U}) \stackrel{\triangle}{=} \sup_{\mathbf{X}} \langle \mathbf{U}, \mathbf{X} \rangle - h(\mathbf{X})$. Based on Fenchel conjugate, the fairness objective can be transformed as variational representation $h_f(\mathbf{p}) = \sup_{\mathbf{u}} \langle \mathbf{p}, \mathbf{u} \rangle - h_f^*(\mathbf{u})$, where $\mathbf{p} = \Delta_s SF(\mathbf{F}) \in \mathbb{R}^{1 \times d_{out}}$ is a predicted probability vector for classification. Furthermore, the original optimization problem is equivalent to

$$\min_{\mathbf{F}} \max_{\mathbf{u}} h_s(\mathbf{F}) + \langle \mathbf{p}, \mathbf{u} \rangle - h_f^*(\mathbf{u}) \tag{2}$$

where $\mathbf{u} \in \mathbb{R}^{1 \times d_{out}}$ and $h_f^*(\cdot)$ is the conjugate function of fairness objective $h_f(\cdot)$.

### 3.2.2 Problem Solution

Motivated by Proximal Alternating Predictor-Corrector (PAPC) (Loris & Verhoeven, 2011; Chen et al., 2013), the min-max optimization problem (2) can be solved by the following fixed-point

---

[4] Such smoothness objetive is the the most common-used one in existing methods (Ma et al., 2021b; Belkin & Niyogi, 2001; Kalofolias, 2016). The various other smoothness objectives could be considered to improve the performance of FMP and we leave it for future work.

equations

$$\begin{cases} \mathbf{F} = \mathbf{F} - \nabla h_s(\mathbf{F}) - \frac{\partial \langle \mathbf{p}, \mathbf{u} \rangle}{\partial \mathbf{F}}, \\ \mathbf{u} = \text{prox}_{h_f^*}\big(\mathbf{u} + \boldsymbol{\Delta}_{\mathbf{s}} SF(\mathbf{F})\big). \end{cases} \quad (3)$$

where $\text{prox}_{h_f^*}(\mathbf{u}) = \arg\min_{\mathbf{y}} ||\mathbf{y} - \mathbf{u}||_F^2 + h_f^*(\mathbf{u})$. Similar to "predictor-corrector" algorithm (Loris & Verhoeven, 2011), we adopt iterative algorithm to find saddle point for min-max optimization problem. Specifically, starting from $(\mathbf{F}^k, \mathbf{u}^k)$, we adopt a gradient descent step on primal variable $\mathbf{F}$ to arrive $(\bar{\mathbf{F}}^{k+1}, \mathbf{u}^k)$ and then followed by a proximal ascent step in the dual variable $\mathbf{u}$. Finally, a gradient descent step on primal variable in point $(\bar{\mathbf{F}}^{k+1}, \mathbf{u}^k)$ to arrive at $(\mathbf{F}^{k+1}, \mathbf{u}^k)$. In short, the iteration can be summarized as

$$\begin{cases} \bar{\mathbf{F}}^{k+1} = \mathbf{F}^k - \gamma \nabla h_s(\mathbf{F}^k) - \gamma \frac{\partial \langle \mathbf{p}, \mathbf{u}^k \rangle}{\partial \mathbf{F}}\Big|_{\mathbf{F}^k}, \\ \mathbf{u}^{k+1} = \text{prox}_{\beta h_f^*}\big(\mathbf{u}^k + \beta \boldsymbol{\Delta}_{\mathbf{s}} SF(\bar{\mathbf{F}}^{k+1})\big), \\ \bar{\mathbf{F}}^{k+1} = \mathbf{F}^k - \gamma \nabla h_s(\mathbf{F}^k) - \gamma \frac{\partial \langle \mathbf{p}, \mathbf{u}^{k+1} \rangle}{\partial \mathbf{F}}\Big|_{\mathbf{F}^k}. \end{cases} \quad (4)$$

where $\gamma$ and $\beta$ are the step size for primal and dual variables. Note that the close-form for $\frac{\partial \langle \mathbf{p}, \mathbf{u} \rangle}{\partial \mathbf{F}} \in \mathbb{R}^{n \times d_{out}}$ and $\text{prox}_{\beta h_f^*}(\cdot)$ are still not clear, we will provide the solution one by one.

**Proximal Operator.** As for the proximal operators, we provide the close-form in the following proposition:

**Proposition 1 (Proximal Operators)** *The proximal operators $prox_{\beta h_f^*}(\mathbf{u})$ satisfies*

$$prox_{\beta h_f^*}(\mathbf{u})_j = sign(\mathbf{u})_j \min\big(|\mathbf{u}_j|, \lambda_f\big), \quad (5)$$

*where $sign(\cdot)$ and $\lambda_f$ are element-wise sign function and hyperparameter for fairness objective. In other words, such proximal operator is element-wise projection onto $l_\infty$ ball with radius $\lambda_f$.*

**FMP Scheme.** Similar to works (Ma et al., 2021b; Liu et al., 2021), choosing $\gamma = \frac{1}{1+\lambda_s}$ and $\beta = \frac{1}{2\gamma}$, we have

$$\mathbf{F}^k - \gamma \nabla h_s(\mathbf{F}^k) = \big((1-\gamma)\mathbf{I} - \gamma\lambda_s\tilde{\mathbf{L}}\big)\mathbf{F}^k + \gamma\mathbf{X}_{trans} = \gamma\mathbf{X}_{trans} + (1-\gamma)\tilde{\mathbf{A}}\mathbf{F}^k, \quad (6)$$

Therefore, we can summarize the proposed FMP as two phases, including propagation with skip connection (Step ❶) and bias mitigation (Steps ❷-❺). For bias mitigation, Step ❷ updates the aggregated node features for fairness objective; Steps ❸ and ❹ aim to learn and "reshape" perturbation vector in probability space, respectively. Step ❺ explicitly mitigate the bias of node features based on gradient descent on primal variable. The mathematical formulation is given as follows:

$$\begin{cases} \mathbf{X}_{agg}^{k+1} = \gamma\mathbf{X}_{trans} + (1-\gamma)\tilde{\mathbf{A}}\mathbf{X}^k, & \text{Step ❶} \\ \bar{\mathbf{F}}^{k+1} = \mathbf{X}_{agg}^{k+1} - \gamma\frac{\partial \langle \mathbf{p}, \mathbf{u}^k \rangle}{\partial \mathbf{F}}\Big|_{\mathbf{F}^k}, & \text{Step ❷} \\ \bar{\mathbf{u}}^{k+1} = \mathbf{u}^k + \beta\boldsymbol{\Delta}_{\mathbf{s}} SF(\bar{\mathbf{F}}^{k+1}), & \text{Step ❸} \\ \mathbf{u}^{k+1} = \min\big(|\bar{\mathbf{u}}^{k+1}|, \lambda_f\big) \cdot sign(\bar{\mathbf{u}}^{k+1}), & \text{Step ❹} \\ \mathbf{F}^{k+1} = \mathbf{X}_{agg}^{k+1} - \gamma\frac{\partial \langle \mathbf{p}, \mathbf{u}^{k+1} \rangle}{\partial \mathbf{F}}\Big|_{\mathbf{F}^k}. & \text{Step ❺} \end{cases}$$

where $\mathbf{X}_{agg}^{k+1}$ represents the node features with normal aggregation and skip connection with the transformed input $\mathbf{X}_{trans}$.

### 3.2.3 GRADIENT COMPUTATION ACCELERATION

The softmax property is also adopted to accelerate the gradient computation. Note that $\mathbf{p} = \boldsymbol{\Delta}_s SF(\mathbf{F})$ and $SF(\cdot)$ represents softmax over column dimension, directly computing the gradient $\frac{\partial \langle \mathbf{p}, \mathbf{u} \rangle}{\partial \mathbf{F}}$ based on chain rule involves the three-dimensional tensor $\frac{\partial \mathbf{p}}{\partial \mathbf{F}}$ with gigantic computation complexity. Instead, we simplify the gradient computation based on the property of softmax function in the following theorem.

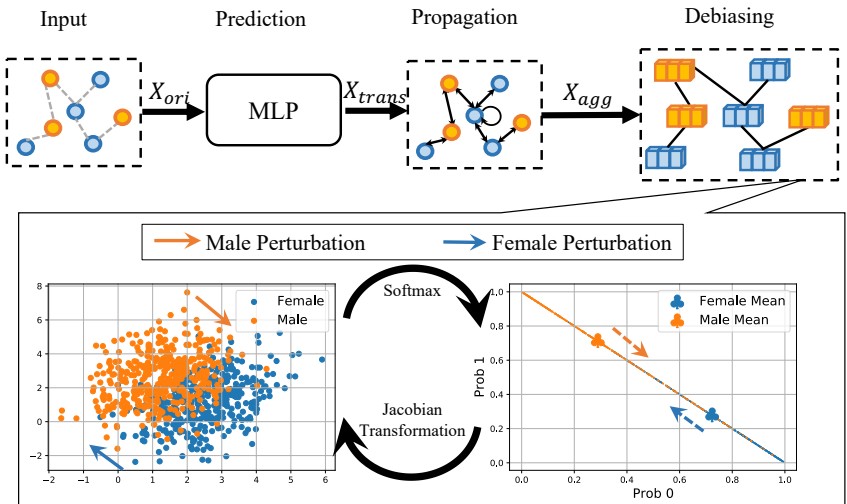

Figure 1: The model pipeline consists of three steps: MLP (feature transformation), propagation with skip connection and debiasing via low-rank perturbation in probability space.

**Theorem 1 (Gradient Computation)** *The gradient over primal variable $\frac{\partial \langle \mathbf{p}, \mathbf{u} \rangle}{\partial \mathbf{F}}$ satisfies*

$$\frac{\partial \langle \mathbf{p}, \mathbf{u} \rangle}{\partial \mathbf{F}} = \mathbf{U}_s \odot SF(\mathbf{F}) - Sum_1(\mathbf{U}_s \odot SF(\mathbf{F}))SF(\mathbf{F}). \tag{7}$$

*where $\mathbf{U}_s \overset{\triangle}{=} \Delta_s^\top \mathbf{u}$, $\odot$ represents element-wise product and $Sum_1(\cdot)$ represents the summation over column dimension with preserved matrix shape.*

### 3.3 DISCUSSION ON FMP

In this section, we provide the interpretation and analyze the *transparency*, *efficiency*, and *effectiveness* of proposed FMP scheme. Specifically, we interprete FMP as two phrases, including conventional aggregation and bias mitigation, and the computation complexity of FMP is lower than backward gradient calculation.

**Interpretation** Note that the gradient of fairness objective over node features $\mathbf{F}$ satisfies $\frac{\partial \langle \mathbf{p}, \mathbf{u} \rangle}{\partial \mathbf{F}} = \frac{\partial \langle \mathbf{p}, \mathbf{u} \rangle}{\partial SF(\mathbf{F})} \frac{\partial SF(\mathbf{F})}{\partial \mathbf{F}}$ and $\frac{\partial \langle \mathbf{p}, \mathbf{u} \rangle}{\partial SF(\mathbf{F})} = \Delta_s^\top \mathbf{u}$, such gradient calculation can be interpreted as three steps: Softmax transformation, perturbation in probability space, and debiasing in representation space. Specifcally, we first map the node representation into probability space via softmax transformation. Subsequently, we calculate the gradient of fairness objective in probability space. It is seen that the perturbation $\Delta_s^\top \mathbf{u}$ actually poses *low-rank* debiasing in probability space, where the nodes with different sensitive attribute embrace opposite perturbation. In other words, *the dual variable $\mathbf{u}$ represents the perturbation direction in probability space.* Finally, the perturbation in probability space will be transformed into representation space via Jacobian transformation $\frac{\partial SF(\mathbf{F})}{\partial \mathbf{F}}$.

**Transparency** The proposed FMP explicitly uses the sensitive attribute information in Steps ❷-❺ during forward propagation. In other words, if we aims to identify the influence of sensitive attributes for FMP, it is sufficient to check the difference between the input and output for debiasing step. It is worth mentioning that the required information for identifying the influence of sensitive attributes are naturally from the forward propagation. However, for the fair model from existing works (e.g, adding regularization and adversarial debiasing), note that the sensitive attribute information is implicitly encoded in the well-trained model weight, the sensitive attribute perturbation inevitably lead to variability of well-trained model weight. Therefore, it is required to retrain the model for probing the influence of sensitive attribute perturbation. The key drawback of these methods is due to encoding the sensitive attributes information into well-trained model weights. From aditors' perspective, it is quite hard to identify the influence of sensitive attributes only given well-trained fair model. Instead,

Table 1: Comparative Results with Baselines on Node Classification.

| Models | Pokec-z | | | Pokec-n | | | NBA | | |
|---|---|---|---|---|---|---|---|---|---|
| | Acc (%) ↑ | $\Delta_{DP}$ (%) ↓ | $\Delta_{EO}$ (%) ↓ | Acc (%) ↑ | $\Delta_{DP}$ (%) ↓ | $\Delta_{EO}$ (%) ↓ | Acc (%) ↑ | $\Delta_{DP}$ (%) ↓ | $\Delta_{EO}$ (%) ↓ |
| MLP | $70.48 \pm 0.77$ | $1.61 \pm 1.29$ | $2.22 \pm 1.01$ | $\mathbf{72.48} \pm 0.26$ | $1.53 \pm 0.89$ | $3.39 \pm 2.37$ | $65.56 \pm 1.62$ | $22.37 \pm 1.87$ | $18.00 \pm 3.52$ |
| GAT | $69.76 \pm 1.30$ | $2.39 \pm 0.62$ | $2.91 \pm 0.97$ | $71.00 \pm 0.48$ | $3.71 \pm 2.15$ | $7.50 \pm 2.88$ | $57.78 \pm 10.65$ | $20.12 \pm 16.18$ | $13.00 \pm 13.37$ |
| GCN | $71.78 \pm 0.37$ | $3.25 \pm 2.35$ | $2.36 \pm 2.09$ | $73.09 \pm 0.28$ | $3.48 \pm 0.47$ | $5.16 \pm 1.38$ | $61.90 \pm 1.00$ | $23.70 \pm 2.74$ | $17.50 \pm 2.63$ |
| SGC | $71.24 \pm 0.46$ | $4.81 \pm 0.30$ | $4.79 \pm 2.27$ | $71.46 \pm 0.41$ | $2.22 \pm 0.29$ | $3.85 \pm 1.63$ | $63.17 \pm 0.63$ | $22.56 \pm 3.94$ | $14.33 \pm 2.16$ |
| APPNP | $66.91 \pm 1.46$ | $3.90 \pm 0.69$ | $5.71 \pm 1.29$ | $69.80 \pm 0.89$ | $1.98 \pm 1.30$ | $4.01 \pm 2.36$ | $63.80 \pm 1.19$ | $26.51 \pm 3.33$ | $20.00 \pm 4.56$ |
| FMP | $\mathbf{70.50} \pm 0.50$ | $\mathbf{0.81} \pm 0.40$ | $\mathbf{1.73} \pm 1.03$ | $72.16 \pm 0.33$ | $\mathbf{0.66} \pm 0.40$ | $\mathbf{1.47} \pm 0.87$ | $\mathbf{73.33} \pm 1.85$ | $\mathbf{18.92} \pm 2.28$ | $\mathbf{13.33} \pm 5.89$ |

our designed FMP explicitly adopts the sensitive attribute information in the forward propagation process, which naturally avoid the dilemma that sensitive attributes are encoded into well-trained model weight. In a nutshell, FMP encompasses with higher transparency since (1) the sensitive attribute is explicitly adopted in forward propagation; (2) It is not necessary to retrain model for probing influence of sensitive attribute.

**Efficiency** FMP is an efficient message passing scheme. The computation complexity for the aggregation (sparse matrix multiplications) is $O(md_{out})$, where $m$ is the number of edges in the graph. For FMP, the extra computation mainly focus on the perturbation calculation, as shown in Theorem 1, with the computation complexity $O(nd_{out})$. The extra computation complexity is negligible in that the number of nodes $n$ is far less than the number of edges $m$ in the real-world graph. Additionally, if directly adopting autograd to calculate the gradient via back propagation, we have to calculate the three-dimensional tensor $\frac{\partial \mathbf{p}}{\partial \mathbf{F}}$ with computation complexity $O(n^2 d_{out})$. In other words, thanks to the softmax property, we achieve an efficient fair message passing scheme.

**Effectiveness** The proposed FMP explicitly achieves graphs smoothness prior and fairness via alternative gradient descent. In other words, the propagation and debiasing forward in a white-box manner and there is not any trainable weight during forwarding phrase. The effectiveness of the proposed FMP is also validated in the experiments on three real-world datasets.

## 4 EXPERIMENTS

In this section, we conduct experiments to validate the effectiveness and efficiency of proposed FMP. We firstly validate that graph data with large sensitive homophily enhances bias in GNNs via synthetic experiments. Moreover, for experiments on real-world datasets, we introduce the experimental settings and then evaluate our proposed FMP compared with several baselines in terms of prediction performance and fairness metrics.

### 4.1 EXPERIMENTAL SETTINGS

**Datasets.** We conduct experiments on real-world datasets Pokec-z, Pokec-n, and NBA (Dai & Wang, 2021). Pokec-z and Pokec-n are sampled, based on province information, from a larger Facebook-like social network Pokec (Takac & Zabovsky, 2012) in Slovakia, where region information is treated as the sensitive attribute and the predicted label is the working field of the users. NBA dataset is extended from a Kaggle dataset [5] consisting of around 400 NBA basketball players. The information of players includes age, nationality, and salary in the 2016-2017 season. The players' link relationships is from Twitter with the official crawling API. The binary nationality (U.S. and overseas player) is adopted as the sensitive attribute and the prediction label is whether the salary is higher than the median.

**Evaluation Metrics.** We adopt accuracy to evaluate the performance of node classification task. As for fairness metric, we adopt two quantitative group fairness metrics to measure the prediction bias. According to works (Louizos et al., 2015; Beutel et al., 2017), we adopt *demographic parity* $\Delta_{DP} = |\mathbb{P}(\hat{y} = 1|s = -1) - \mathbb{P}(\hat{y} = 1|s = 1)|$ and *equal opportunity* $\Delta_{EO} = |\mathbb{P}(\hat{y} = 1|s =$

---
[5]https://www.kaggle.com/noahgift/social-power-nba

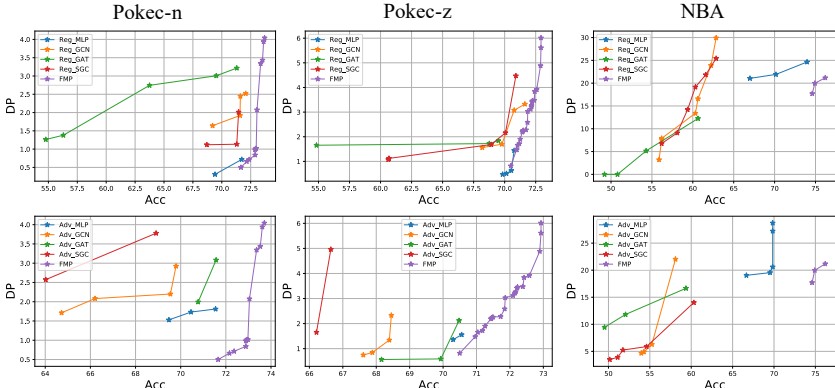

Figure 2: DP and Acc trade-off performance on three real-world datasets compared with adding regularization (Top) and adversarial debiasing (Bottom). The trade-off curve more close to right bottom corner means better trade-off performance [6]

$-1, y = 1) - \mathbb{P}(\hat{y} = 1|s = 1, y = 1)|$, where $y$ and $\hat{y}$ represent the ground-truth label and predicted label, respectively.

**Baselines.** We compare our proposed FMP with representative GNNs, such as GCN (Kipf & Welling, 2017), GAT (Veličković et al., 2018), SGC (Wu et al., 2019), and APPNP (Klicpera et al., 2019), and MLP. For all models, we train 2 layers neural networks with 64 hidden units for 300 epochs. Additionally, We also compare adversarial debiasing and adding demographic regularization methods to show the effectiveness of the proposed method.

**Implementation Details.** We run the experiments 5 times and report the average performance for each method. We adopt Adam optimizer with $0.001$ learning rate and $10^{-5}$ weight decay for all models. For adversarial debiasing, we adopt train classifier and adversary with 70 and 30 epochs, respectively. The hyperparameter for adversary loss is tuned in $\{0.0, 1.0, 2.0, 5.0, 8.0, 10.0, 20.0, 30.0\}$. For adding regularization, we adopt the hyperparameter set $\{0.0, 1.0, 2.0, 5.0, 8.0, 10.0, 20.0, 50.0, 80.0, 100.0\}$.

### 4.2 EXPERIMENTAL RESULTS

**Comparison with existng GNNs.** The accuracy, demographic parity and equal opportunity metrics of proposed FMP for Pokec-z, Pokec-n, NBA dataset are shown in Table 1 compared with MLP, GAT, GCN, SGC and APPNP. The detailed statistical information for these three dataset is shown in Table 3. From these results, we can obtain the following observations:

- Many existing GNNs underperorm MLP model on all three datasets in terms of fairness metric. For instance, the demographic parity of MLP is lower than GAT, GCN, SGC and APPNP by $32.64\%$, $50.46\%$, $66.53\%$ and $58.72\%$ on Pokec-z dataset. The higher prediction bias comes from the aggregation within the-same-sensitive-attribute nodes and topology bias in graph data.
- Our proposed FMP consistently achieve lowest prediction bias in terms of demographic parity and equal opportunity on all datasets. Specifically, FMP reduces demographic parity by $49.69\%$, $56.86\%$ and $5.97\%$ compared with the lowest bias among all baselines in Pokec-z, Pokec-n, and NBA dataset. Meanwhile, our proposed FMP achieves the best accuracy in NBA dataset, and comparable accuracy in Pokec-z and Pokec-n datasets. In a nutshell, proposed FMP can effectively mitigate prediction bias while preserving the prediction performance.

**Comparison with adversarial debiasing and regularization.** To validate the effectiveness of proposed FMP, we also show the prediction performance and fairness metric trade-off compared with fairness-boosting methods, including adversarial debiasing (Fisher et al., 2020) and adding regularization (Chuang & Mroueh, 2020). Similar to (Louppe et al., 2017), the output of GNNs is the input of adversary and the goal of adversary is to predict the node sensitive attribute. We also adopt several backbones for these two methods, including MLP, GCN, GAT and SGC. We randomly split

50%/25%/25% for training, validation and test dataset. Figure 2 shows the pareto optimality curve for all methods, where right-bottom corner point represents the ideal performance (highest accuracy and lowest prediction bias). From the results, we list the following observations as follows:

- Our proposed FMP can achieve better DP-Acc trade-off compared with adversarial debiasing and adding regularization for many GNNs and MLP. Such observation validates the effectiveness of the key idea in FMP: aggregation first and then debiasing. Additionally, FMP can reduce demographic parity with negligible performance cost due to transparent and efficient debiasing.

- Message passing in GNNs does matter. For adding regularization or adversarial debiasing, different GNNs embrace huge distinctionwhich implies that appropriate message passing manner potentially leads to better trade-off performance. Additionally, many GNNs underperforms MLP in low label homophily coefficient dataset, such as NBA. The rationale is that aggregation may not always bring benefit in terms of accuracy when the neighbors have low probability with the same label.

## 5    RELATED WORKS

**Graph neural networks.**    GNNs generalizing neural networks for graph data have already been shown great success for various real-world applications. There are two streams in GNNs model design, i.e., spectral-based and spatial-based. Spectral-based GNNs provide graph convolution definition based on graph theory, which is utilized in GNN layers together with feature transformation (Bruna et al., 2013; Defferrard et al., 2016; Henaff et al., 2015). Graph convolutional networks (GCN) (Kipf & Welling, 2017) simplifies spectral-based GNN model into spatial aggregation scheme. Since then, many spatial-based GNNs variant are developed to update node representation via aggregating its neighbors' information, including graph attention network (GAT) (Veličković et al., 2018), GraphSAGE (Hamilton et al., 2017), SGC (Wu et al., 2019), APPNP (Klicpera et al., 2019), et al (Gao et al., 2018; Monti et al., 2017). Graph signal denoising is another perspective to understand GNNs. Recently, there are several works show that GCN is equivalent to the first-order approximation for graph denoising with Laplacian regularization (Henaff et al., 2015; Zhao & Akoglu, 2019). The unified optimization framework are provided to unify many existing message passing schemes (Ma et al., 2021b; Zhu et al., 2021a).

**Fairness-aware learning on graphs.**    Many works have been developed to achieve fairness in machine learning community (Jiang et al., 2022; Chuang & Mroueh, 2020; Zhang et al., 2018; Du et al., 2021; Yurochkin & Sun, 2020; Creager et al., 2019; Feldman et al., 2015). A pilot study on fair node representation learning is developed based on random walk (Rahman et al., 2019). Additionally, adversarial debiasing is adopt to learn fair prediction or node representation so that the well-trained adversary can not predict the sensitvie attribute based on node representation or prediction (Dai & Wang, 2021; Bose & Hamilton, 2019; Fisher et al., 2020). A Bayesian approach is developed to learn fair node representation via encoding sensitive information in prior distribution in (Buyl & De Bie, 2020). Work (Ma et al., 2021a) develops a PAC-Bayesian analysis to connect subgroup generalziation with accuracy parity. (Laclau et al., 2021; Li et al., 2021) aim to mitigate prediction bias for link prediction. Fairness-aware graph contrastive learning are proposed in (Agarwal et al., 2021; Köse & Shen, 2021). However, aforementioned works ignore the requirement of transparency in fairness. In this work, we develop an efficient and transparent fair message passing scheme explicitly rendering sensitive attribute usage.

## 6    CONCLUSION

In this work, we consider fairness problem from a new perspective, named transparency, i.e., the influence of sensitive attribute should be easily probed to public. We point out that existing fairness model in graph lack of transparency due to implicitly encoding sensitive attribute information in well-trained model weight. Additionally, integrated in a unified optimization framework, we develop an efficient, effective and transparent FMP to learn fair node representations while preserving prediction performance. Experimental results on real-world datasets demonstrate the effectiveness and efficiency compared with state-of-the-art baselines in node classification tasks.

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

## A  NOTATIONS

Table 2: Table of Notations

| Notations | Description |
|---|---|
| $|\mathcal{E}|$ | The number of edges |
| $n$ | The number of nodes |
| $d$ | The number of node feature dimensions |
| $d_{out}$ | The number of node classes |
| $\Delta_{\mathbf{s}} \in \mathbb{R}^{1 \times n}$ | The sensitive attribute incident vector |
| $\epsilon_{label}$ | Label homophily coefficient |
| $\epsilon_{sens}$ | Sensitive homophily coefficient |
| $\mathbf{X}_{ori} \in \mathbb{R}^{n \times d}$ | The input node attributes matrix |
| $\mathbf{A} \in \mathbb{R}^{n \times n}$ | The adjacency matrix |
| $\hat{\mathbf{A}} \in \mathbb{R}^{n \times n}$ | The adjacency matrix with self-loop |
| $\tilde{\mathbf{A}} \in \mathbb{R}^{n \times n}$ | The normalized adjacency matrix with self-loop |
| $\mathbf{L} \in \mathbb{R}^{n \times n}$ | The Laplacian matrix |
| $\mathbf{X}_{trans} \in \mathbb{R}^{n \times d_{out}}$ | The output node features for feature transformation |
| $\mathbf{F}_{agg} \in \mathbb{R}^{n \times d_{out}}$ | The aggregated node features after propagation |
| $\mathbf{F} \in \mathbb{R}^{n \times d_{out}}$ | The learned node features considering graph smoothness and fairness |
| $\mathbf{u} \in \mathbb{R}^{1 \times d_{out}}$ | The permutation direction in feature representation space |
| $h^*(\cdot)$ | Fenchel conjugate function of $h(\cdot)$ |
| $||\mathbf{X}||_F, ||\mathbf{X}||_1$ | The Frobenius norm and $l_1$ norm of matrix $\mathbf{X}$ |
| $\lambda_f, \lambda_s$ | Hyperparameter for fairness and graph smoothness objectives |

## B  PROOF ON FAIRNESS OBJECTIVE

The fairness objective can be shown as the average prediction probability difference as follows:

$$
\begin{aligned}
\left(\Delta_s SF(\mathbf{F}))\right)_j &= \left[\frac{\mathbb{1}_{>0}(\mathbf{s})}{||\mathbb{1}_{>0}(\mathbf{s})||_1} - \frac{\mathbb{1}_{>0}(-\mathbf{s})}{||\mathbb{1}_{>0}(-\mathbf{s})||_1}\right]\left(SF(\mathbf{F})\right)_{:,j} \\
&= \frac{\sum_{\mathbf{s}_i=1} \hat{P}(y_i = j|\mathbf{X})}{||\mathbb{1}_{>0}(\mathbf{s})||_1} - \frac{\sum_{\mathbf{s}_i=-1} \hat{P}(y_i = j|\mathbf{X})}{||\mathbb{1}_{>0}(-\mathbf{s})||_1} \\
&= \hat{P}(y_i = j|\mathbf{s}_i = 1, \mathbf{X}) - \hat{P}(y_i = j|\mathbf{s}_i = -1, \mathbf{X}).
\end{aligned}
$$

## C  PROOF OF THEOREM 1

Before providing in-depth analysis on the gradient computation, we first introduce the softmax function derivative property in the following lemma:

**Lemma 1** *For the softmax function with $N$-dimensional vector input $\mathbf{y} = SF(\mathbf{x}) : \mathbb{R}^{1 \times N} \longrightarrow \mathbb{R}^{1 \times N}$, where $y_j = \frac{e^{\mathbf{x}_j}}{\sum_{k=1}^{N} e^{\mathbf{x}_k}}$ for $\forall j \in \{1, 2, \cdots, N\}$, the derivative $N \times N$ Jocobian matrix is defined by $[\frac{\partial \mathbf{y}}{\partial \mathbf{x}}]_{ij} = \frac{\partial \mathbf{y}_i}{\partial \mathbf{x}_j}$. Additionally, Jocobian matrix satisfies $\frac{\partial \mathbf{y}}{\partial \mathbf{x}} = \mathbf{y}\mathbf{I}_N - \mathbf{y}^{\top}\mathbf{y}$, where $\mathbf{I}_N$ represents $N \times N$ identity matrix and $\top$ denotes the transpose operation for vector or matrix.*

**Proof:**  *Considering the gradient $\frac{\partial \mathbf{y}_i}{\partial \mathbf{x}_j}$ for arbitrary $ij$, according to quotient and chain rule of derivatives, we have*

$$
\frac{\partial \mathbf{y}_i}{\partial \mathbf{x}_j} = \frac{e^{\mathbf{x}_i} \sum_{k=1}^{N} e^{\mathbf{x}_k} - e^{\mathbf{x}_i + \mathbf{x}_j}}{\left(\sum_{k=1}^{N} e^{\mathbf{x}_k}\right)^2} = \frac{e^{\mathbf{x}_i}}{\sum_{k=1}^{N} e^{\mathbf{x}_k}} \cdot \frac{\sum_{k=1}^{N} e^{\mathbf{x}_k} - e^{\mathbf{x}_i}}{\sum_{k=1}^{N} e^{\mathbf{x}_k}} = \mathbf{y}_i(1 - \mathbf{y}_j), \tag{8}
$$

*Similarly, for arbitrary $i \neq j$, the gradient is given by*

$$
\frac{\partial \mathbf{y}_i}{\partial \mathbf{x}_j} = \frac{e^{\mathbf{x}_i}}{\sum_{k=1}^{N} e^{\mathbf{x}_k}} \cdot \frac{-e^{\mathbf{x}_i}}{\sum_{k=1}^{N} e^{\mathbf{x}_k}} = -\mathbf{y}_i\mathbf{y}_j. \tag{9}
$$

*Combining these two cases, it is easy to verify the Jocobian matrix satisfies $\frac{\partial \mathbf{y}}{\partial \mathbf{x}} = \mathbf{y}\mathbf{I}_N - \mathbf{y}^\top\mathbf{y}$.* □

Arming with the derivative property of softmax function, we further investigate the gradient $\frac{\partial\langle\mathbf{p},\mathbf{u}\rangle}{\partial\mathbf{F}}$, where $\mathbf{p} = \mathbf{\Delta}_s SF(\mathbf{F}) \in \mathbb{R}^{1\times d_{out}}$ and $SF(\cdot)$ and $\mathbf{u} \in \mathbb{R}^{1\times d_{out}}$ is independent with $\mathbf{F} \in \mathbb{R}^{n\times d_{out}}$.

Considering softmax function $SF(\mathbf{x}) \in \mathbb{R}^{n\times d}$ is row-wise adopted in node representation matrix, the gradient satisfies $\frac{\partial SF(\mathbf{F})_i}{\partial\mathbf{F}_j} = \mathbf{0}_{d_{out}\times d_{out}}$ for $i \neq j$. Note that the inner product $\langle\mathbf{p},\mathbf{u}\rangle = \sum_{k=1}^{d_{out}}\mathbf{p}_k\mathbf{u}_k$, it is easy the obtain the gradient $[\frac{\partial\langle\mathbf{p},\mathbf{u}\rangle}{\partial\mathbf{F}}]_{ij} = \sum_{k=1}^{d_{out}}\frac{\partial\mathbf{p}_k}{\partial\mathbf{F}_{ij}}\mathbf{u}_k$.

To simply the current notation, we denote $\tilde{\mathbf{F}} \overset{\triangle}{=} SF(\mathbf{F})$. According to chain rule of derivative, we have

$$\frac{\partial\mathbf{p}_k}{\partial\mathbf{F}_{ij}} = \sum_{t=1}^{d_{out}}\frac{\partial\mathbf{p}_k}{\partial\tilde{\mathbf{F}}_{tk}}\frac{\partial\tilde{\mathbf{F}}_{tk}}{\partial\mathbf{F}_{ij}} = \sum_{t=1}^{d_{out}}\mathbf{\Delta}_{s,t}\frac{\partial\tilde{\mathbf{F}}_{tk}}{\partial\mathbf{F}_{ij}} \overset{(a)}{=} \mathbf{\Delta}_{s,i}\frac{\partial\tilde{\mathbf{F}}_{ik}}{\partial\mathbf{F}_{ij}} \overset{(b)}{=} \mathbf{\Delta}_{s,i}\tilde{\mathbf{F}}_{ik}[\delta_{kj} - \tilde{\mathbf{F}}_{ij}], \tag{10}$$

where $\delta_{kj}$ is Dirac function (equals 1 only if $k = j$, otherwise 0;), equality (a) holds since softmax function is row-wise operation, and equality (b) is based on Lemma 1. Furthermore, we can obtain the gradient of fairness objective w.r.t. node presentation as follows:

$$[\frac{\partial\langle\mathbf{p},\mathbf{u}\rangle}{\partial\mathbf{F}}]_{ij} = \sum_{k=1}^{d_{out}}\frac{\partial\mathbf{p}_k}{\partial\mathbf{F}_{ij}}\mathbf{u}_k = \sum_{k=1}^{d_{out}}\mathbf{\Delta}_{s,i}\tilde{\mathbf{F}}_{ik}[\delta_{kj} - \tilde{\mathbf{F}}_{ij}]\mathbf{u}_k = \mathbf{\Delta}_{s,i}\tilde{\mathbf{F}}_{ij}\mathbf{u}_j - \mathbf{\Delta}_{s,i}\tilde{\mathbf{F}}_{ij}\sum_{k=1}^{d_{out}}\tilde{\mathbf{F}}_{ik}\mathbf{u}_k. \tag{11}$$

Therefore, the matrix formulation is given by

$$\frac{\partial\langle\mathbf{p},\mathbf{u}\rangle}{\partial\mathbf{F}} = \mathbf{U}_s \odot SF(\mathbf{F}) - \text{Sum}_1(\mathbf{U}_s \odot SF(\mathbf{F}))SF(\mathbf{F}). \tag{12}$$

where $\mathbf{U}_s \overset{\triangle}{=} \mathbf{\Delta}_s^\top\mathbf{u} \in \mathbb{R}^{n\times d_{out}}$ and $\text{Sum}_1(\cdot)$ represents the summation over column dimension with preserved matrix shape. Therefore, the computation complexity for gradient $\frac{\partial\langle\mathbf{p},\mathbf{u}\rangle}{\partial\mathbf{F}}$ is $O(nd_{out})$.

## D    PROOF OF PROPOSITION 1

We firstly show the conjugate function for general norm function $f(\mathbf{x}) = \lambda||\mathbf{x}||$, where $\mathbf{x} \in \mathbf{R}^{1\times d_{out}}$. The conjugate function of $f(\mathbf{x})$ satisfies

$$f^*(\mathbf{y}) = \begin{cases} 0, & ||\mathbf{x}||_* \leq \lambda, \\ +\infty, & ||\mathbf{x}||_* > \lambda. \end{cases} \tag{13}$$

where $||\mathbf{x}||_*$ is dual norm of the original norm $||\mathbf{x}||$, defined as $||\mathbf{y}||_* = \max_{||\mathbf{x}||\leq 1}\mathbf{y}^\top\mathbf{x}$. Considering the conjugate function definition $f^*(\mathbf{y}) = \max_{\mathbf{x}}\mathbf{y}^\top\mathbf{x} - \lambda||\mathbf{x}||$ the analysis can be divided as the following two cases:

❶ If $||\mathbf{y}||_* \leq \lambda$, according to the definition of dual norm, we have $\mathbf{y}^\top\mathbf{x} \leq ||\mathbf{x}||||\mathbf{y}||_* \leq \lambda||\mathbf{x}||$ for $\forall||\mathbf{x}||$, where the equality holds if and only if $||\mathbf{x}|| = 0$. Hence, it is easy to obtain $f^*(\mathbf{y}) = \max_{\mathbf{x}}\mathbf{y}^\top\mathbf{x} - \lambda||\mathbf{x}|| = 0$.

❷ If $||\mathbf{y}||_* > \lambda$, note that the dual norm $||\mathbf{y}||_* = \max_{||\mathbf{x}||\leq 1}\mathbf{y}^\top\mathbf{x} > \lambda$, there exists variables $\hat{\mathbf{x}}$ so that $||\hat{\mathbf{x}}|| \leq 1$ and $\hat{\mathbf{x}}^\top\mathbf{y} < \lambda$. Therefore, for any constant $t$, we have $f^*(\mathbf{y}) \geq \mathbf{y}^\top(t\mathbf{x}) - \lambda||t\mathbf{x}|| = t(\mathbf{y}^\top\mathbf{x} - \lambda||\mathbf{x}||) \overset{t\to\infty}{\longrightarrow} \infty$.

Based on aforementioned two cases, it is easy to get the conjugate function for $l_1$ norm (the dual norm is $l_\infty$), i.e., the conjugate function for $h_f(\mathbf{x}) = \lambda||\mathbf{x}||_1$ is given by

$$h_f^*(\mathbf{y}) = \begin{cases} 0, & ||\mathbf{x}||_\infty \leq \lambda, \\ +\infty, & ||\mathbf{x}||_\infty > \lambda. \end{cases} \tag{14}$$

Given the conjugate function $h_f^*(\cdot)$, we further investigate the proximal operators $\text{prox}_{h_f^*}$. Note that

$$\text{prox}_{h_f^*}(\mathbf{u}) = \arg\min_{\mathbf{y}}||\mathbf{y} - \mathbf{u}||_F^2 + h_f^*(\mathbf{u}) = \arg\min_{||\mathbf{y}||_\infty\leq\lambda_f}||\mathbf{y} - \mathbf{u}||_F^2 = \arg\min_{\substack{\mathbf{y}_j\leq\lambda_f \\ \forall j\in[d_{out}]}}\sum_{j=1}^{d_{out}}|\mathbf{y}_j -$$

$\mathbf{u}_j|^2$, the proximal operator problem can be decomposed as element-wise sub-problem, i.e.,

$$\text{prox}_{h_f^*}(\mathbf{u})_j = \arg \min_{\mathbf{y}_j \leq \lambda_f} |\mathbf{y}_j - \mathbf{u}_j|^2 = sign(\mathbf{u}_j) \min(|\mathbf{u}_j|, \lambda_f)$$

which completes the proof.

## E    TRAINING ALGORITHMS

We summarize the training algorithm for FMP and provide the pseudo codes in Algorithm 1.

---
**Algorithm 1:** FMP Training Algorithm
---
**Input**    : Graph dataset $=(\mathbf{X}, \mathbf{A}, \mathbf{Y})$; The total epochs $T$; Hyperparameters $\lambda_s$ and $\lambda_f$.
**Output** : The well-trained FMP model.
1 **for** *epoch from 1 to T* **do**
2      Conduct feature transformation using MLP;
3      Conduct propagation and debiasing as steps ❶-❺;
4      Calculate the cross entropy loss for node classification task;
5      Conduct back propagation step to update model weight;
6 **end**
---

## F    DATASET STATISTICS

For fair comparison with previous work, we perform the node classification task on three real-world dataset, including Pokec-n, Pokec-z, and NBA. The data statistical information on three real-world dataset is provided in Table 3. It is seen that the sensitive homophily are even higher than label homophily coefficient among three real-world dataset, which validates that the real-world dataset are usually with large topology bias.

Table 3: Statistical Information on Datasets

| Dataset | # Nodes | # Node Features | # Edges | # Training Labels | # Training Sens |
|---|---|---|---|---|---|
| Pokec-n | 66569 | 265 | 1034094 | 4398 | 500 |
| Pokec-z | 67796 | 276 | 1235916 | 5131 | 500 |
| NBA | 403 | 95 | 21242 | 156 | 246 |

## G    MORE DISCUSSION ON GNNS AS GRAPH SIGNAL DENOISING

In this section, we provide more examples to show many existing GNNs can be interpreted as graph signal denoising problem, including GAT and APPNP.

**GAT.**    Feature aggregation in GAT applies the normalized attention coefficient to compute a linear combination of neighbor's features as $\mathbf{X}_{agg,i} = \sum_{j \in \mathcal{N}(i)} \alpha_{ij} \mathbf{X}_{trans,j}$, where $\alpha_{ij} = softmax_j(e_{ij})$, $e_{ij} = \text{LeakyReLU}(\mathbf{X}_{trans,i}^{\top} \mathbf{w}_i + \mathbf{X}_{trans,j}^{\top} \mathbf{w}_j)$, and $\mathbf{w}_i$ and $\mathbf{w}_j$ are learnable column vectors. Prior study (Ma et al., 2021b) demonstrates that one-step gradient descent with adaptive step size $\frac{1}{\sum_{j \in \mathcal{N}(i)} (c_i + c_j)}$ for the following objective problem:

**PPNP / APPNP.**    Feature aggregation in PPNP and APPNP adopt the aggregation rules as $\mathbf{X}_{agg} = \alpha \Big( \mathbf{I} - (1 - \alpha)\tilde{\mathbf{A}} \Big)^{-1} \mathbf{X}_{trans}$ and $\mathbf{X}_{agg}^{k+1} = (1 - \alpha)\tilde{\mathbf{A}}\mathbf{X}_{agg}^k + \alpha\mathbf{X}_{trans}$. It is shown that they are equivalent to the exact solution and one gradient descent step with stepsize $\frac{\alpha}{2}$ to minimize the following objective problem:

$$\min_{\mathbf{F}} ||\mathbf{F} - \mathbf{X}_{trans}||_F^2 + (\frac{1}{\alpha} - 1)tr\Big( \mathbf{F}^{\top}(\mathbf{I} - \tilde{\mathbf{A}})\mathbf{F} \Big).$$

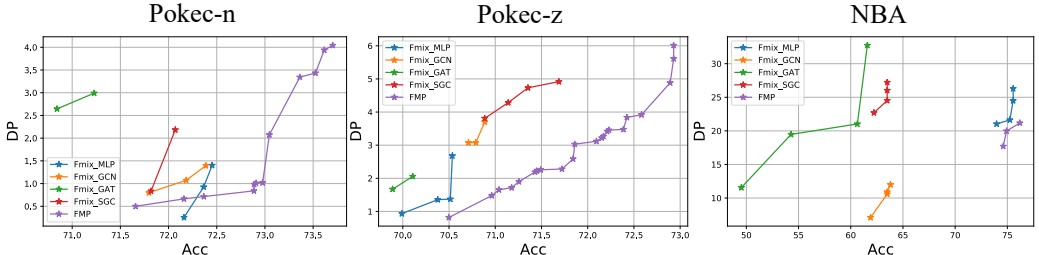

Figure 3: DP and Acc trade-off performance on three real-world datasets compared with (manifold) Fair Mixup.

## H MORE EXPERIMENTAL RESULTS

### H.1 MORE EXPERIMENTAL SETTING DETAILS

In FMP implementation, we first use 2 layers MLP with 64 hidden units and the output dimension for MLP is 2. We also stack 2 layers for propagation and debiasing steps, where there is not any trainable model parameters. As for the model training, we adopt cross entropy loss function with 300 epochs. We also adopt Adam optimizer with $0.001$ learning rate and $1 \times 10^5$ weight decay for all models. The hyperprameters for FMP is $\lambda_f = \{0, 5, 10, 15, 20, 30, 100\}$ and $\lambda_s = \{0, 0.01, 0.1, 0.5, 1.0, 2.0, 3, 5, 10, 15, 20\}$ .

### H.2 COMPARISON WITH FAIR MIXUP

We also implement Fair mixup (Chuang & Mroueh, 2021) as the additional baseline for different GNN backbones in Figure 3. Note that input fair mixup requires calculating model prediction for mixed input batch, it is non-trivial to adopt input fair mixup in our experiments (node classification task) since forward propagation in GNN aggregates information from neighborhoods while the neighborhood information for the mixed input batch is missing. Therefore, we adopt manifold fair mixup for the logit layer (the previous layers contain aggregation step) in our experiments. Experimental results show that our method can still achieve better accuracy-fairness tradeoff performance on three datasets.

### H.3 SENSITIVE ATTRIBUTE INFLUENCE PROBE

As for lending fairness perceptron, it represents the influence of sensitive attributes could be identified. For example, our proposed FMP includes three steps, i.e., transformation, aggregation and debiasing, where the sensitive attribute explicitly adopted in debiasing step. If we aims to identify the influence of sensitive attributes for FMP, it is sufficient to check the difference between the input and output for debiasing step. It is worth mentioning that the required information for identifying the influence of sensitive attributes are naturally from the forward propagation. Additionally, if we aim to identify the influence of sensitive attributes for existing methods (e.g, adding regularization and adversarial debiasing), the well-trained fair model is insufficient and we need additional vanilla (unfair) model without using any sensitive attribute information. In other words, these methods require model retraining with sensitive attribute removement and thus much more resources for sensitive attributes influence auditing. The key drawback of these methods is due to encoding the sensitive attributes information into well-trained model weights. From auditors' perspective, it is quite hard to identify the influence of sensitive attributes only given well-trained fair model. Instead, our designed FMP explicitly adopts the sensitive attribute information in the forward propagation process, which naturally avoid the dilemma that sensitive attributes are encoded into well-trained model weight.

Figure 4 shows the visualization results for training with/without (left/right) sensitive attribute for FMP and several baselines (with GCN backbones) across three real-world datasets. From the visualization results, we observe that all methods with sensitive attribute information achieve better fairness since the logit layer representation for different sensitive attributes are mixed with each other. Therefore, it is hard to identify the sensitive attribute based on the representation and thus leads to higher fairness results. The key different is that the results for training with/without (left/right)

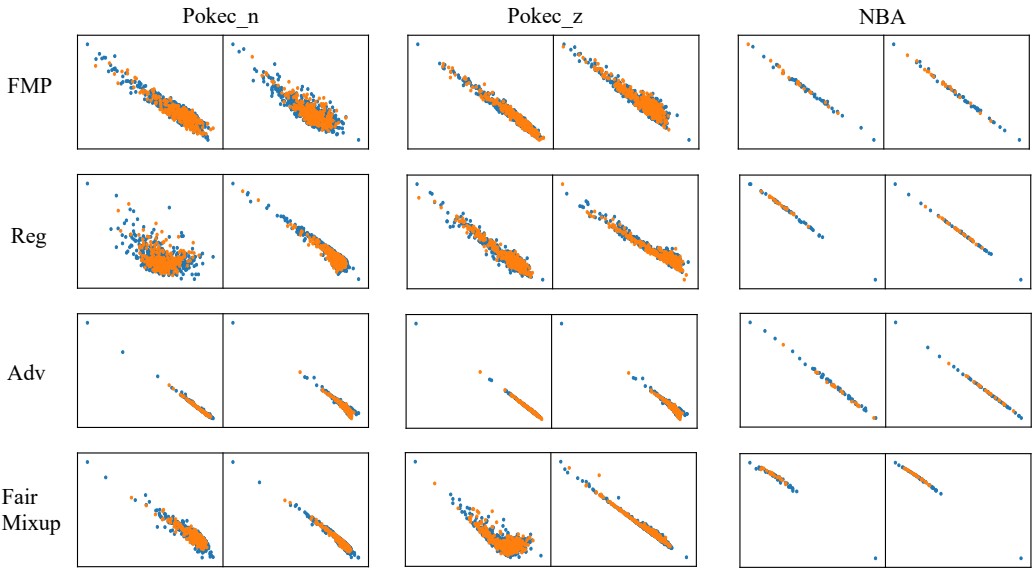

Figure 4: The visualization of node representation for training with/without (left/right) sensitive attribute for FMP and several baselines across three real-world dataset. The data point with different colors represents different sensitive attributes.

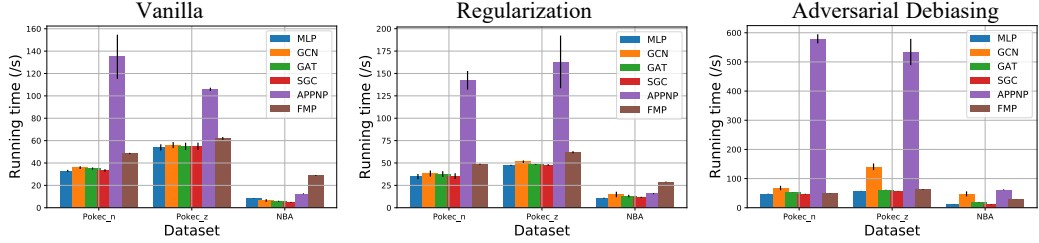

Figure 5: The running time comparison.

sensitive attribute in FMP can both obtained through the forward propagation, while the other baseline methods requiring model retraining to probe the influence of sensitive attributes.

## H.4 RUNNING TIME COMPARISON

We provide running time comparison in Figure 5 for our proposed FMP and other baselines, including vanilla, regularization and adversarial debiasing on many backbones (MLP, GCN, GAT, SGC, and APPNP). To achieve fair comparison, we adopt the same Adam optimizer with 200 epochs with 5 running times. We list several observations as follows:

- The running time of proposed FMP is very efficient for large-scale dataset. Specifically, for vanilla method, the running time of FMP is higher than most lighten backbone MLP with $46.97\%$ and $15.03\%$ time overhead on Pokec-n and Poken-z dataset, respectively. Compared with the most time-consumption APPNP, the running time of FMP is lower with $64.07\%$ and $41.45\%$ time overhead on Pokec-n and Poken-z dataset, respectively.

- The regularization method achieves almost the same running time compared with vanilla method on all backbones. For example, GCN with regularization encompasses higher running time with $6.41\%$ time overhead compared with vanilla method. Adversarial debiasing is extremely time-consuming. For example, GCN with adversarial debiasing encompasses higher running time with $88.58\%$ time overhead compared with vanilla method.

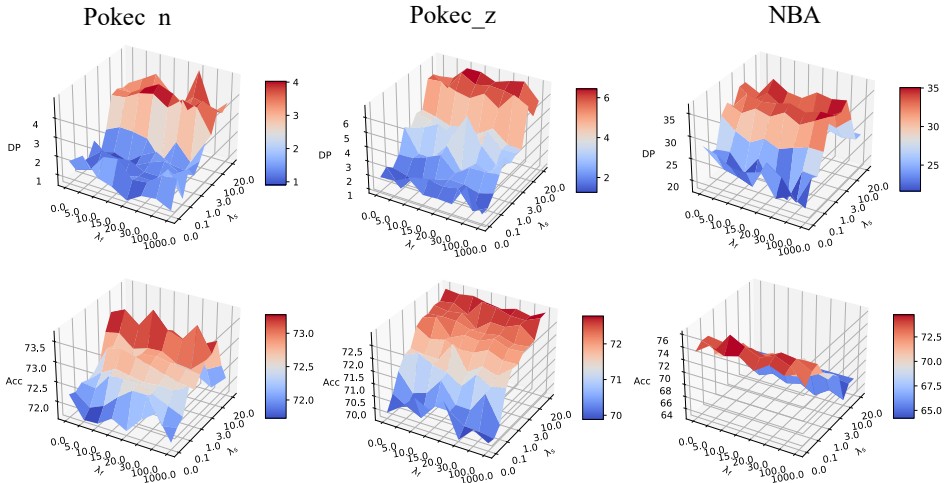

Figure 6: Hyperparameter study on fairness and smoothness hyperparameter for demographic parity and Accuracy.

## H.5 HYPERPARAMETER STUDY

We provide hyperparameter study for further investigation on fairness and smoothness hyperparmeter on prediction and fairness performance on three datasets. Specifically, we tune hyperparameters as $\lambda_f = \{0.0, 5.0, 10.0, 15.0, 20.0, 30.0, 100.0, 1000.0\}$ and $\lambda_s = \{0.0, 0.1, 0.5, 1.0, 3.0, 5.0, 10.0, 15.0, 20.0\}$. From the results in Figure 6, we can make the following observations:

- The accuracy and demographic parity are extremely sensitive to smoothness hyperparameter. It is seen that, for Pokec-n and Pokec-z datasets (NBA), larger smoothness hyperparameter usually leads to higher (lower) accuracy with higher prediction bias. The rationale is that, only for graph data with high label homophily coefficient, GCN-like aggregation with skip connection is beneficial. Otherwise, the neighbor's node representation with different label will mislead representation update.

- The appropriate fairness hyperparameter leads to better fairness and prediction performance tradeoff. The reason is that fairness hyperparameter determinates the perturbation vector update step size in probability space. Only appropriate step size can lead to better perturbation vector update.

## I BROADER SOCIAL IMPACT AND LIMITATIONS

Transparency is an advanced property in fairness domain and poses huge challenging for research and industrial. Many existing works mainly rely on specific fairness metric to evaluate the prediction bias. The transparency may stimulates maintainers and auditors of machine learning system to rethink the fairness evaluation/auditing. Only achieving fair model with lower bias for specific fairness metric is insufficient. The maintainers should also consider how to leverage the influence of sensitive attribute for auditors. The transparency may lead maintainers pay more effects in improving the transparency of fair model and could be helpful to convince the auditors. The limitations of this work are that it requests sensitive information in the inference stage.

