# OpenReview forum: "Fair Graph Message Passing with Transparency"
_ICLR.cc/2023/Conference — Submitted to ICLR 2023_

### Official Review · Reviewer_hAa7 · 2022-10-19

**Confidence:** 4
**Correctness:** 3
**Technical Novelty And Significance:** 3
**Empirical Novelty And Significance:** 3
**Recommendation:** 6

**Clarity, Quality, Novelty And Reproducibility:**

The paper has various typos and some flaws based on the definitions provided. The hyperparameter details along with the empirical evidence presented suggest that the results are likely to be reproducible. The technical novelty of the work is reasonable in that they propose to study the fairness problem through the angle of “transparency” and suggest an interesting solution.


**Strength And Weaknesses:**

1. The results provided in table 1 and figure 2 are indeed strong. The proposed FMP scheme clearly outperforms all the baselines.
2. The runtime comparison results are quite convincing. The proposed method indeed holds value in that the runtime vs performance tradeoff is good.
3. There are various typos in the text writeup as well as in some equations. I strongly encourage the authors to thoroughly read the paper again.
4. If the authors are talking about the proximal operator commonly used in optimization, then the definition of proximal operator in the first paragraph of page 5 is incorrect. It should be ${h^*}_{f}(y)$. Based on this, the proof of proposition 1 seems to have some flaws. Also, the projection in eq 5 seems “into” the ball rather than “onto”.
5. The authors mention that - “directly computing the gradient of the fairness term makes the closed-form gradient complicated since the gradient of $l_1$ norm involves the sign of elements in the vector”. This is the rationale behind the proposed bi-level optimization framework. Perhaps, some experiment to compare their method against the standard $l_1$ norm gradient computation, will be helpful to verify the efficacy of their method.


**Summary Of The Paper:**

The authors propose to study the problem of fair node classification on graphs through the angle of “transparency”. They characterize transparency as the ability to verify models - (i) by quantifying the influence of the sensitive attribute in model prediction and (ii) obtaining such quantification using the trained model and test data samples. They point out critical flaws in the existing graph fairness methods in that such models encode the fairness property implicitly in the trained model weights and therefore don’t trivially adhere to the second aforementioned property. They propose a two-step framework that accounts for both graph aggregation and debiasing, to optimize for eq 1. To ensure the aggregation procedure over the graph, they leverage the connection between aggregation in certain GNNs and one-step gradient descent to minimize a particular function stated in section 2.2. To ensure fairness, they leverage the Fenchel conjugate of the fairness objective function. Eventually, this converts the problem into a Bi-level optimization which is solved using alternating optimization given by the fixed-point equations in eq 3. A closed-form operation for the proximal operator is provided which finally converts the optimization procedure into the proposed FMP scheme provided on page 5. A gradient acceleration trick is also suggested to compute the partial derivative of $<p,u>$ w.r.t $F$. Experiments on popular graph fairness datasets suggest that the proposed scheme is indeed better as compared to GNN baselines (without any fairness objectives) as well as to previously proposed SOTA graph fairness methods.


**Summary Of The Review:**

Based on the questions, comments and concerns raised in the previous sections, I lean towards weak acceptance of the work.

---

> ### Author Response · Authors · 2022-11-19
> **Response to Reviewer hAa7**
>
> We thank the reviewer for the constructive comments and appreciate the reviewer for recognition of the novelty of our work. Please see the revised version at [https://anonymous.4open.science/r/FMP-AD84/ICLR23_FMP.pdf](https://anonymous.4open.science/r/FMP-AD84/ICLR23_FMP.pdf)
>
> Q1: There are various typos in the text writeup as well as in some equations. I strongly encourage the authors to thoroughly read the paper again.
>
> A1: We carefully go through the manuscript and submit a revised version.
>
> Q2: If the authors are talking about the proximal operator commonly used in optimization, then the definition of the proximal operator in the first paragraph of page 5 is incorrect. It should be $h^*_{f}(\mathbf{y})$. Based on this, the proof of proposition 1 seems to have some flaws. Also, the projection in eq 5 seems “into” the ball rather than “onto”.
>
> A2: We have revised typos the definition of the proximal operator on page 5 and Appendix D accordingly. The proof in Appendix D still holds since the item $h^{*}_{f}(\mathbf{y})$ is $0$ on the last paragraph of page 14.
>
> Q3: The authors mention that - “directly computing the gradient of the fairness term makes the closed-form gradient complicated since the gradient of norm involves the sign of elements in the vector”. This is the rationale behind the proposed bi-level optimization framework. Perhaps, some experiment to compare their method against the standard norm gradient computation will be helpful to verify the efficacy of their method.
>
> A3: Thanks for pointing out this baseline directly using the gradient of fairness term (named "ML1"). We have added the results of this baseline in Table. 1, Figures. 2 and 3. It is seen that the proposed FMP can achieve better tradeoff performance than the baseline "ML1".

---

### Official Review · Reviewer_RsB6 · 2022-10-24

**Confidence:** 4
**Clarity, Quality, Novelty And Reproducibility:** 1.{Evaluation
**Correctness:** 4
**Technical Novelty And Significance:** 3
**Empirical Novelty And Significance:** 3
**Recommendation:** 5

**Strength And Weaknesses:**

Strength:

1. Consider the fairness problem from a new perspective, and develop a transparent method named FMP to learn fair node representations while preserving prediction performance, which has generalization.

2. The code is available, and the results can be reproduced.

3. The model is effective according to to reduce gradient computational complexity with theoretical and empirical validation.

Weakness:

1. This paper compares some classic methods such as APPAP, SGC, etc., but lacks some comparisons of SOTA methods.

2. If more general datasets could be compared such as Cora, citeseer, etc. It would be better.

3. The scale of NBA datasets is small, with only 400 nodes. The performance improvement on NBA datasets cannot provide enough experimental support.

4. Limited performance improvement. In table 1, the experimental results of marking black are not the best performance. For example, under datasets pokec-z, although the other two indicators improved, GCN has the best AUC.  If possible, please offer the statistical significance test of your model.

**Summary Of The Paper:**

This paper considers the fairness problem from a new perspective, named transparency, i.e., the influence of sensitive attribute should be easily probed by the public, and propose a novel method named FMP to achieve fairness with transparency via using sensitive attribute information in message passing. To accelerate the training process, the paper proposed an accelerated method to reduce gradient computational complexity with theoretical and empirical validation.

**Summary Of The Review:**

It is good to work with great novelty, but experiments should be more sufficient.

---

> ### Author Response · Authors · 2022-11-19
> **Response to Reviewer RsB6**
>
> We thank the reviewer for the constructive comments. Please see the revised version at [https://anonymous.4open.science/r/FMP-AD84/ICLR23_FMP.pdf](https://anonymous.4open.science/r/FMP-AD84/ICLR23_FMP.pdf)
>
> Q1: This paper compares some classic methods such as APPAP, SGC, etc., but lacks some comparisons of SOTA methods.
>
> A1: We would like to clarify that the proposed FMP is the first work to achieve fair prediction in graph learning from the **model backbone perspective**. Many previous works achieve fair prediction in the graph via either graph topology/node feature preprocessing or fair training strategies, such as adding regularization, adversarial debiasing, or Fair mixup. To address the reviewer's concern, we have added one more GNNs backbone JKNet, ML1(directly using the gradient of optimization (1) in the forward propagation), and two more datasets. Additionally, Our FMP explores fair graph learning from a model perspective via explicitly using sensitive attributes in forward propagation. In other words, there is **not any data pre-processing or post-processing parts and using cross-entropy loss** can even achieve fair prediction. In the experiments, our goal is to validate that achieving fair prediction from a model perspective is tractable, instead of achieving SOTA tradeoff performance. We believe that the experimental results are **sufficient to support the effectiveness** of our proposed method.
>
> Q2: If more general datasets could be compared such as Cora, citeseer, etc. It would be better.
>
> A2:  The graph datasets Cora and citeseer **cannot** be adopted in fairness graph learning since the node sensitive attributes (such as race, male) are **missing**. For fairness graph learning. we do notice that there are some **constructed** graph datasets, including Recidivism graph, and Credit defaulter graph, adopted in [1], where the graph topology for these datasets is constructed based on the **similarity of node features** instead of from ground truth. To do a more comprehensive study, we add experimental results on Credit graph, Recidivism graph.
>
> [1] Agarwal, Chirag, Himabindu Lakkaraju, and Marinka Zitnik. "Towards a unified framework for fair and stable graph representation learning." Uncertainty in Artificial Intelligence. PMLR, 2021.
>
> Q3: The scale of NBA datasets is small, with only 400 nodes. The performance improvement on NBA datasets cannot provide enough experimental support.
>
> A3: The NBA dataset is common-used in fair graph learning [2] although it is small. To the best of my knowledge, Pokec-n, Pokec-z, and NBA are the most common-used datasets in fair graph learning. Due to the sensitivity of collecting sensitive attributes for each node (person), there is no large-scale dataset for fair graph learning. Additionally, there are constructed graph datasets from tabular data adopted in research work based on feature similarity. We conduct experiments on two additional datasets. Please see Appendix~H.6 for more details.
>
>
> [2] Dai, Enyan, and Suhang Wang. "Say no to the discrimination: Learning fair graph neural networks with limited sensitive attribute information." Proceedings of the 14th ACM International Conference on Web Search and Data Mining. 2021.
>
> Q4: Limited performance improvement. In table 1, the experimental results of marking black are not the best performance. For example, under datasets pokec-z, although the other two indicators improved, GCN has the best AUC. If possible, please offer the statistical significance test of your model.
>
> A4: We would like to clarify that our contributions are mainly focusing on achieving fairness in graph learning from a **new model perspective**. Many previous works on fair graph learning either rely on data preprocessing or fair training strategy. Our FMP explores fair graph learning from a model perspective via explicitly using sensitive attributes in forward propagation. In other words, there is not any data preprocessing part, and using cross-entropy loss can even achieve fair prediction. In the experiments, we believe that the experimental results are sufficient to support the effectiveness of our proposed method. The statistical significance test is not usually used in computer science publications since the improvement of most work is not significantly better than baselines in the sense of the statistical test.

---

> ### Author Response · Authors · 2022-11-29
> **Looking forward to your response**
>
> Dear Reviewer RsB6,
>
> Thank you again for the constructive and valuable comments. As we are nearing the end of the discussion phase, we would like to know if our response has addressed your questions. If you have any further questions, we would be more than happy to address them.
>
> Thanks, Authors

---

### Official Review · Reviewer_QgmT · 2022-11-03

**Confidence:** 3
**Correctness:** 3
**Technical Novelty And Significance:** 2
**Empirical Novelty And Significance:** 2
**Recommendation:** 3

**Clarity, Quality, Novelty And Reproducibility:**

- Paper is not particularly clear in its presentation. Reasoning is sometimes hard to follow and English writing is not perfect. Grammar and syntax errors can be found several times per page.


**Strength And Weaknesses:**

_Strengths_

- The FMP method outperforms the chosen baselines;
- Literature on fair representation learning specifically for GNNs is scarce, therefore the paper tackles an arguably under-explored field;


_Weaknesses_

- The concept of _transparency_ is somewhat arbitrary and not fully clear. The authors claim a "formal definition" in page 1, but what follows is arguably subjective.
    - Is it just that a model is "transparent" if it uses sensitive attribute information in its decision-making process? Usually the opposite condition (not explicitly using the sensitive attribute) is targeted in methods from the literature. It's known that sensitive attribute information is redundantly encoded on any sufficiently-rich real-world dataset. Using this information explicitly or not is not novel, and its not clear which choice is best (it actually depends heavily on the specific real-world task and context). A significative part of the paper is dedicated to this concept of transparency which seems quite trivial.
    - In the context of deep learning, even by explicitly using sensitive attribute information in the forward pass, this in no way brings us closer to understanding exactly _how_ the sensitive attribute information is used to produce predictions in a human-understandable manner.

- Baselines are limited. Arguably the simplest fairness intervention you can use is that of [Hardt et al. (2016)](https://proceedings.neurips.cc/paper/2016/file/9d2682367c3935defcb1f9e247a97c0d-Paper.pdf). Adding this thresholding-based post-processing step to the GNN baselines would be an easy-to-obtain baseline for comparison with the more complex method presented here.
    - Usually there is a clear fairness-performance trade-off in every task/dataset. If FMP achieves the best fairness, I find it hard to believe that it would achieve the best performance as well. This signals baselines weren't properly selected (e.g., using a simple GNN + thresholding post-processing or other fairness interventions).
- Experiments were repeated 5 times and the final results averaged, which is a positive. However variance is not shown in Figure 2, and it is not used to compute whether differences in fairness/performance are statistically significative.
    - For instance, the NBA dataset is quite small, and large variance between experiment runs is expected;

_Other notes_
- Table 1, the bold on the FMP method is not correct as it does not achieve the highest performance;
    - Bold on EO of FMP on the NBA dataset is also not correct;
- The formula given for DP and EO is the absolute difference in probabilities; as such the y-axis of the plots in Figure 2 should be between 0 and 1; if it is in percentage than this should be explained in the legend or caption;
- Figure 2 caption has a footnote "6" that I cannot find in the paper;

**Summary Of The Paper:**

The paper follows two separate trains of thought.
The first is on advocating for the idea of transparency w.r.t. how a given model produces fair predictions. The authors define transparency as explicitly using the sensitive attribute in forward propagation.
The second, and main point, is the development of a Fair Message Passing (FMP) method. According to the paper's results, FMP seems to outperform the chosen baselines on every point of the fairness-accuracy trade-off.

**Summary Of The Review:**

Overall, some non-trivial modifications are needed for the paper to be ready for publication, namely: honing down focus and presentation efforts on the FMP method, adding well-known baselines from the fairness literature, and correcting English writing.

---

> ### Author Response · Authors · 2022-11-19
> **Response to Reviewer QgmT [Part 1/2:  More discussion on transparency and baselines]**
>
> We thank the reviewer for the constructive comments and appreciate the reviewer for the recognition of the novelty of our work. Please see the revised version at [https://anonymous.4open.science/r/FMP-AD84/ICLR23_FMP.pdf](https://anonymous.4open.science/r/FMP-AD84/ICLR23_FMP.pdf)
>
> Q1: The concept of transparency is somewhat arbitrary and not fully clear. The authors claim a "formal definition" on page 1, but what follows is arguably subjective. Is it just that a model is "transparent" if it uses sensitive attribute information in its decision-making process? In the context of deep learning, even by explicitly using sensitive attribute information in the forward pass, this in no way brings us closer to understanding exactly how the sensitive attribute information is used to produce predictions in a human-understandable manner.
>
> A1: In our work, transparency represents that the auditors/maintainers can **identify the influence of sensitive attributes from the model itself without any other external methods**. In other words, transparency represents whether the influence of sensitive attributes can be identified from the model itself. Transparency could be regarded as a "yes" or "no" (binary) problem in this paper. As a comparison, the transparency is similar to the model intrinsic explainability. Intrinsic explanation refers to machine learning models that are considered explainable due to their simple structure, such as short decision trees or sparse linear models. Unlike explainable methods relying on an extenal model for interpretation, intrinsic explanation derives from the model itself.
>
> The designed FMP is naturally transparent while many fair methods are not since the sensitive attribute information is encoded in well-trained model parameters. It is not **necessary** that a model is "transparent" if it uses sensitive attribute information in its decision-making process. As we mentioned above, transparency requires that the influence of sensitive attributes can be identified from the model itself. For graph data, if the model utilizes input features, graph topology, and sensitive attributes in a highly **entangled** way (such as iteratively using this information), the influence of sensitive attributes is not easy to obtain only from the model itself. Instead, the proposed FMP **separately** adopts input features, graph topology and sensitive attributes in different stages (feature transformation, aggregation, and debiasing) of Figure 1. Therefore, the influence of sensitive attributes can be easily identified through the output of different stages. As for human understandability, we don't claim that transparency is human-understandable. Instead, transparency can provide the influence of sensitive attributes from the model itself (just like intrinsic explainable can provide an explaination from the model itseelf without any external models/methods).
>
> Q2: Baselines are limited. Arguably the simplest fairness intervention you can use is that of Hardt et al. (2016). Adding this thresholding-based post-processing step to the GNN baselines would be an easy-to-obtain baseline for comparison with the more complex method presented here.
>
> A2: We would like to clarify that our main contribution is to design a fair message-passing scheme to achieve fair prediction with **transparency** in graph data. The proposed FMP is naturally with transparency since the sensitive attribute is adopted during forward propagation in a rather **structural/separated** way. Additionally, FMP achieves fair prediction from a new (model backbone) perspective, while most of the previous works achieve fair prediction relying on **either data pre-processing in-processing (fair training strategy), or post-processing**. FMP achieves fair prediction with cross-entropy loss **without** any pre- or post-processing. Our goal is to validate that FMP can achieve fair prediction instead of achieving state-of-the-art performance.
> In this work, we adopt adding regularization, adversarial debiasing, and fair mixup across different GNNs backbone (GCN, GAT, and SGC) as the baseline for three graph datasets. In the revised version, we have added one more GNNs backbone JKNet, ML1 (directly use the gradient of optimization (1) in the forward propagation), and two more datasets. We believe that the current experiments can support the effectiveness of FMP. Please see Appendix~H.6 for more details.

---

> ### Author Response · Authors · 2022-11-19
> **Response to Reviewer QgmT [Part 2/2:  Minor comments]**
>
> Q3: The variance is not shown in Figure 2, and it is not used to compute whether differences in fairness/performance are statistically significant.
>
> A3: We mainly aim to compare the tradeoff performance for different methods in Figure 2. Considering the variance of the fairness metric is relatively large, we don't show the variance in Figure. 2 since it might distract readers to compare different methods. As for statistical significance, it is rarely adopted to compare different methods since the improvements over baseline are insufficient to be statistically significant. We usually compared the average performance metric (such as accuracy) with running multiple times.
>
> Q4: Table 1, the bold on the FMP method is not correct as it does not achieve the highest performance; Figure 2 caption has a footnote "6" that I cannot find in the paper;
>
> A4: We have revised the bold in Table 1. We also revised the caption in Figure 2.
>
> Q5: The formula given for DP and EO is the absolute difference in probabilities; as such the y-axis of the plots in Figure 2 should be between 0 and 1; if it is in percentage then this should be explained in the legend or caption;
>
> A5: We add more description on the value with the percentage at x-axis and y-axis in Figures 2 and 3.

---

> ### Author Response · Authors · 2022-11-29
> **Looking forward to your response**
>
> Dear Reviewer QgmT,
>
> Thank you again for the constructive and valuable comments. As we are nearing the end of the discussion phase, we would like to know if our response has addressed your questions. If you have any further questions, we would be more than happy to address them.
>
> Thanks,
> Authors

---

### Official Review · Reviewer_Sogu · 2022-11-04

**Confidence:** 3
**Correctness:** 3
**Technical Novelty And Significance:** 3
**Empirical Novelty And Significance:** 2
**Recommendation:** 5

**Clarity, Quality, Novelty And Reproducibility:**

* The paper is relatively clear and readable
* My understanding is that it is technically/mathematically a novel and non-trivial extension of prior work, but I am not entirely convinced by the motivation and am not sure if the solution proposed by the paper is necessary.
* The authors provide link to their source code, so I believe the work should be reproducible to a reasonable extent.

**Details Of Ethics Concerns:**

I have raised my concerns for possible, unintended breach of the double-blind review policy privately to the PC, SACs and ACs.

**Strength And Weaknesses:**


## Strengths

* The paper is clearly written and well-structured, which facilitates its understanding and readability.
* It concerns fairness in GNNs, which in my opinion is still relatively underexplored topic.
* The informally proposed "transparency in fairness" is conceptually nice and can potentially inspire further studies in the intersection between fairness and explainability.
* To the best of my understanding, the paper extends prior work in intuitive yet mathematically non-trivial way.

## Weaknesses, Questions and Concerns

My main concerns are about the motivation of this work and how it relates to the experimental evaluation. I also have questions about some of the technical details.

### Experiments

1. The paper suggests that the main experimental setup is similar to that of (Dai & Wang, 2021). If this is the case, then FairGCN and FairGAT (Table 3, Dai & Wang, 2021) seem to outperform FMP (Table 1 of this work) in terms of fairness while the accuracy stays similar or slightly lower. The difference in the results is especially large for the NBA dataset. This comparison is entirely missing from the current version of the paper. If the experimental setup in (Dai & Wang, 2021) differs from that of FMP, it is not made clear how, in what way and why.
2. In general, the cited references for the adversarial debiasing (Fisher et al., 2020) and regularization (Chuang & Mroueh, 2020) methods, which FMP is compared to, seem to not consider GNN fairness, i.e., they have been applied to other datasets and domains. I would rather see comparison to fairness aware methods which are more relevant and have been applied to the GNN fairness domain before. Obvious candidates for that are the aforementioned FairGCN and FairGAT from (Dai & Wang, 2021), but the authors also cite other GNN fairness works in the Related Works section.
3. According to the Introduction, a primary motivation of this work is "to make the process of achieving fair model via sensitive attribute informations white-box" such that "the maintainers and auditors both get benefits from model transparency". Moreover, the paper claims that "chasing explainability can help experts understand how the model provide prediction and convince users". Yet, there is no experimental evaluation how "transparent" FMP actually is, what information about the model it elucidates and how useful and convincing FMP is for the users or model designers.

### Motivation

4. The paper provides a "formal statement on transparency in fairness" in the Introduction, but to me it is rather high-level and informal. I am not aware of a formal mathematical framework or metric that evaluates how transparent a given model is.
5. A selling point of FMP is that it "uncovers how sensitive attributes influence final prediction" and that this is not possible for prior black-box models which "implicitly encode the sensitive attribute information in the well-trained model weight via backward propagation". However, there have been tools such as GNNExplainer (Ying et al., NeurIPS 2019), PGExplainer (Luo et al., NeurIPS 2020), etc., which aim in generating explanations for GNNs. Given the connection to explainability (as mentioned in the paper), I would be interested to know if these tools can be repurposed in the context of "Transparency in fairness" proposed by the authors, but discussion of that is also missing.
6. According to the introduction, "The biased node representation largely limits the application of GNNs in many high-stake tasks", but the provided citations (Mehrabi et al., 2021, Suresh & Guttag, 2019) do not make it clear how GNNs are currently being used in such tasks. In particular, it is not clear what kind of bias the authors aim to mitigate until the reader reaches the experimental section. Is it topology bias, is it representation bias, does the bias comes from the fact that the sensitive attributes are encoded/embedded/included in the input features or the learnt representations? I would suggest to make this more explicit and bring it up somewhere earlier in the paper.

### Technical Details

7. Without reading prior works (I referred to (Ma et al., 2021b) for a general background), it is not clear that the model solves the optimization problem in Eq. (1) for every GNN layer (to the best of my understanding). It is not entirely obvious what $\mathbf{F}$ denotes and at times sections 2 and 3 can be a bit confusing in sense that it is not clear what is plugged when and where in the model.
8. The authors experiment with GNNs with maximum 2 layers. If my assumption from above is correct that Eq. (1) is solved for every GNN layer, it is not obvious to me why the fairness term $h_f$ should be integrated in every layer of the GNN and not only in the last one which provides the predictions. If we talk about solving Eq. (1) for the intermediate layers, I don't understand why we would take the softmax $SF(\mathbf{F})$ of them. There seems to be some possible confusion arising here for me.
9. To the best of my understanding, the FMP method described in Section 3 is only concerned with the _aggregation_ step of the GNN model. However, hypothetically, it should be possible that some bias and unfairness can also arise from the feature transformation steps, but these are not discussed or mentioned in the paper. Thus, it is hard for me to judge the generality and the applicability of FMP.
10. In particular, the end of Section 3 says that "the propagation and debiasing forward in a white-box manner and there is not any trainable weight during forwarding phrase" and I don't understand why this is the case (e.g., if there is feature transformation involved, see $\mathbf{W}$ in Algorithm 1 of (Ma et al., 2021b)).
11. Section 3.2 says that "directly computing the gradient of the fairness term makes the closed-form gradient complicated since the gradient of $l_1$ norm involves the sign of elements in the vector". The gradient of the $l_1$ norm can be tractably computed and I don't entirely understand why the closed-form is absolutely required. E.g., how does it affect the optimization problem if we also try to optimize it with gradient descent (esp. given that $\text{sign}(\mathbf{u})$ is used in Eq. (5))?

### Minor

Some typos, the optimization problem for GAT in App. G seems to be missing and there are some other minor issues related to the paper readability.

**Summary Of The Paper:**

The paper proposes Fair Message Passing (FMP). It builds on Graph Signal Denoising, e.g., as described in (Ma et al., 2021b). The methods described in previous works are extended by adding an additional fairness term ($h_f$ in Eq. (1)) to the optimization problem that governs the feature aggregation step of the GNN. An efficient, iterative algorithm for solving this optimization problem is derived. The main motivation and promise of FMP is to explicitly uncover how sensitive attributes influence the model predictions and explicitly render sensitive attribute usage in the forward computation of the GNN. The authors dub this notion "transparency in fairness". FMP is compared to popular (fairness unaware) GNN baselines as well as adversarial debiasing and demographic regularization methods on 3 datasets (also used in prior work): Pokec-n, Pokec-z, and NBA.

**Summary Of The Review:**

In summary, it seems to me that the paper presents some technical contributions, but I don't find them very well-motivated and am not sure if the evaluation is convincing and thorough enough. I also have minor concerns about some elements of the paper exposition (esp. in Section 3).

---

> ### Author Response · Authors · 2022-11-19
> **Response to Reviewer Sogu [Part 1/3: More discussion on transprency]**
>
> We thank the reviewer for the constructive comments and appreciate the reviewer for the recognition of the novelty of our work. Please see the revised version at [https://anonymous.4open.science/r/FMP-AD84/ICLR23_FMP.pdf](https://anonymous.4open.science/r/FMP-AD84/ICLR23_FMP.pdf)
>
> Q1: The paper suggests that the main experimental setup is similar to that of (Dai & Wang, 2021). If this is the case, then FairGCN and FairGAT  This comparison with FairGCN and FairGAT is entirely missing from the current version of the paper. If the experimental setup in (Dai & Wang, 2021) differs from that of FMP, it is not made clear how, in what way and why.
>
> A1: We believe that the research problem in FairGNN is **different** from ours. We aim to design **fair message passing algorithm** from GNN backbone perspective, while FairGNN considers learning fair GNNs with limited sensitive attribute information from a loss perspective. There are two major differences. First, we focus on fair GNNs backbone design. Instead, FairGNN designs a fair loss to achieve fair prediction. The other difference is that FairGNN considers the case where **limited sensitive attribute information** is available. Specifically, FairGNN uses a GNN-based sensitive attribute estimator to address the lack of sensitive attribute information problem. In our work, we consider a **different case** that all sensitive attribute information is available. Although it is possible to extend our methods to solve limited sensitive information problems and compare them with FairGNN, we believe it is out of the scope of this paper. We add more discussion on FairGNN in future work. Please see Appendix~I for more discussions.
>
> Q2: In general, the cited references for the adversarial debiasing (Fisher et al., 2020) and regularization (Chuang & Mroueh, 2020) methods, which FMP is compared to, seem to not consider GNN fairness, i.e., they have been applied to other datasets and domains.
>
> A2: We would like to clarify that our contributions are mainly focusing on achieving fairness in graph learning from a **new model perspective**. Many previous works on fair graph learning either rely on data preprocessing or fair training strategy. Our FMP explores fair graph learning from a model perspective via explicitly using the sensitive attribute in forward propagation. In other words, there is **not any data pre-processing or post-processing parts and using cross-entropy loss** can even achieve fair prediction. In the experiments, our goal is to validate that achieving fair prediction from a model perspective is tractable, instead of achieving SOTA tradeoff performance. We believe that the experimental results are **sufficient to support the effectiveness** of our proposed method. For the adversarial debiasing (Fisher et al., 2020) and regularization (Chuang & Mroueh, 2020) methods, the such fair loss can be adopted for many data domains. We also conduct experiments on Fair mixup in Appendix H.2.
>
> Q3: There is no experimental evaluation how "transparent" FMP actually is, what information about the model it elucidates and how useful and convincing FMP is for the users or model designers.
>
> A3: In our work, transparency represents that the auditors/maintainers can **identify the influence of sensitive attributes** from **model itself without any other external methods**. In other words, transparency represents whether the influence of sensitive attributes can be identified from the model itself. The transparency could be regarded as a **"yes" or "no"** (binary) problem in this paper. The designed FMP is naturally transparent while many fair methods are not since the sensitive attribute information is encoded in well-trained model parameters. As a comparison, the transparency is **similar** to model intrinsic explainability. Intrinsic explanation refers to machine learning models that are considered explainable due to their simple structure, such as short decision trees or sparse linear models. Unlike explainable methods relying on an extenal model for interpretation, intrinsic explanation derives from the model itself. In summary, FMP is the first GNNs with transparency, which can probe the influence of sensitive attributes only from model propagation with any external models/methods.

---

> ### Author Response · Authors · 2022-11-19
> **Response to Reviewer Sogu [Part 2/3: More discussion on transprency and FMP]**
>
> Q4: The paper provides a "formal statement on transparency in fairness" in the Introduction, but to me it is rather high-level and informal. I am not aware of a formal mathematical framework or metric that evaluates how transparent a given model is. What's the connection with GNN explaination methods, such as GNNExplainer (Ying et al., NeurIPS 2019), PGExplainer (Luo et al., NeurIPS 2020).
>
> A4: Similar to model intrinsic explainability, transparency is essentially **binary**. If the influence of sensitive attributes can be inferred only from the model itself, then we call such a model transparent. In the model interpretability field, if a model (such as decision tree) is intrinsically interpretable, how the model makes the decision is understandable for humans. For the model without intrinsic interpretability, external methods are necessary to develop to help humans understand how the model makes predictions. In this work, FMP is designed with transparency (like the model with intrinsic interpretability). Therefore, it is not necessary to adopt an external method to explain model predictions.
>
> Q5: It is not clear what kind of bias the authors aim to mitigate until the reader reaches the experimental section. Is it topology bias, is it representation bias, does the bias comes from the fact that the sensitive attributes are encoded/embedded/included in the input features or the learnt representations? I would suggest to make this more explicit and bring it up somewhere earlier in the paper.
>
> A5: In this paper, we aim to achieve fair prediction. Topology bias is one of bias from graph data, which leads to bias prediction. We will make it clear in the revised version.
>
> Q6: Without reading prior works (I referred to (Ma et al., 2021b) for a general background), it is not clear that the model solves the optimization problem in Eq. (1) for every GNN layer (to the best of my understanding). It is not entirely obvious what denotes and at times sections 2 and 3 can be a bit confusing in the sense that it is not clear what is plugged when and where in the model.
>
> A6: In previous work (Ma et al., 2021b), a general and universal framework is developed to understand aggregation operations in GNNs. Building on top of this framework, we formulate the optimization problem in Eq. (1) to achieve fair message-passing operation. We will make it clear in the revised version.
>
> Q7: The authors experiment with GNNs with maximum 2 layers. If my assumption from above is correct that Eq. (1) is solved for every GNN layer, it is not obvious to me why the fairness term should be integrated in every layer of the GNN and not only in the last one which provides the predictions. If we talk about solving Eq. (1) for the intermediate layers, I don't understand why we would take the softmax of them. There seems to be some possible confusion arising here for me.
>
> A7: Sorry for the misunderstanding. We would like to clarify that our work is based on the framework proposed by (Ma et al., 2021b), which provides a **universal understanding** for many GNNs **aggregations** from the optimization perspective. Specifically, given input node features/representation, this framework aims to find optimal output node features/representation with specific objectives, such as smoothness. In this way, the input and output are with **the same shape** (like aggregation operation). In our work, we first use MLP model to conduct feature transformation and then use fair message passing to achieve fair aggregation as the final logit. In other words, the feature transformation and fair message passing are **separated** in FMP. Subsequently, we adopt softmax to predict for the node classification task, We will make the designed FMP structure clear in the revised version.

---

> ### Author Response · Authors · 2022-11-19
> **Response to Reviewer Sogu [Part 3/3: More discussion on transprency and FMP]**
>
> Q8: To the best of my understanding, the FMP method described in Section 3 is only concerned with the aggregation step of the GNN model. However, hypothetically, it should be possible that some bias and unfairness can also arise from the feature transformation steps, but these are not discussed or mentioned in the paper. Thus, it is hard for me to judge the generality and applicability of FMP.
>
> A8: As we mentioned in A7, the feature transformation is conducted first. Therefore, the input for fair message passing is the output of MLP. We agree that the feature transformation leads to biased presentation. For fair message passing, there is no feature transformation. Only aggregation and debising steps (no trainable weights) are involved after feature transformation. The designed aggregation and debiasing steps are **operation-level**, we integrate feature transformation, aggregation, and debiasing steps into FMP. Therefore, we believe that FMP is general.
>
> Q9: Section 3.2 says that "directly computing the gradient of the fairness term makes the closed-form gradient complicated since the gradient of $l_1$ norm involves the sign of elements in the vector". The gradient of the norm can be tractably computed and I don't entirely understand why the closed-form is absolutely required. E.g., how does it affect the optimization problem if we also try to optimize it with gradient descent (esp. given that $sign(\mathbf{u})$ is used in Eq. (5))?
>
> A9: The close-form gradient of the optimization problem is needed since we aim to **incorporate this part into model backbone design**. Motivated by work [A], we adopt the “predictor-corrector" algorithm to solve the optimization problem with **per iteration low computation complexity and convergence guarantee**. Additionally, we add one more baseline to directly adopt the gradient of the optimization problem. Experimental results demonstrated that FMP can achieve better performance.  We will revise our paper accordingly.
>
> [A] Ignace Loris and Caroline Verhoeven. On a generalization of the iterative soft-thresholding algorithm
> for the case of non-separable penalty. Inverse Problems, 27(12):125007, 2011.

---

> ### Author Response · Authors · 2022-11-29
> **Looking forward to your response**
>
> Dear Reviewer Sogu,
>
> Thank you again for the constructive and valuable comments. As we are nearing the end of the discussion phase, we would like to know if our response has addressed your questions. If you have any further questions, we would be more than happy to address them.
>
> Thanks,
> Authors

---

> > ### Comment · Reviewer_Sogu · 2022-12-06
> > **Response to the authors**
> >
> > Dear Authors,
> >
> > Thank you for the time and effort you have put to address my questions.
> >
> > While I understand and acknowledge that your research problem is different from that in FairGNN, I still believe that a loss function regularization benchmark (similar to what FairGNN does) would be useful, especially given that the GNN fairness literature is still scarce and both works make use of the same datasets. Moreover, direct comparison of the results suggests that approach like FairGNN can be on par (or sometimes dramatically improve the fairness metrics for the NBA dataset) with FMP.
> >
> > While achieving SOTA results is clearly not a sufficient ground for rejection, in my opinion, the current version of the paper still fails to demonstrate what is interpretable and/or transparent about FMP, both in terms of paper clarity and qualitative analysis. For example, in p. 1 of the intro transparency is described as "How and if the sensitive attribute information influence fair model prediction" but in the rebuttal the authors say that they "first use MLP model to conduct feature transformation and then use fair message passing to achieve fair aggregation as the final logit". Assuming the sensitive attribute information is part of the features, after this first MLP step, it becomes exceedingly difficult to analyze how and if the sensitive attribute information influences the fair model prediction (because all features get bundled together in the intermediate transformation/representation).
> >
> > Due to the above reasons, I stand by my original score.

---

> > > ### Author Response · Authors · 2022-12-07
> > > **Response to Reviewer Sogu [Part 1/2: comparison with regularization]**
> > >
> > > Thanks for the constructive comments and detailed response. We would like to address your concerns. (We also revise our paper based on your comments at https://anonymous.4open.science/r/FMP-AD84/ICLR23_FMP.pdf )
> > >
> > > Q1: I still believe that a loss function regularization benchmark (similar to what FairGNN does) would be useful, especially given that the GNN fairness literature is still scarce and both works make use of the same datasets. Moreover, a direct comparison of the results suggests that approach like FairGNN can be on par (or sometimes dramatically improve the fairness metrics for the NBA dataset) with FMP.
> > >
> > > A1: We agree that the loss function regularization benchmark is useful with many GNNs and MLP. Compared to the regularization methods, we highlight the advantages of FMP as follows:
> > > - Transparency: For a well-trained fair model, FMP can identify the influence of sensitive attribute on model prediction without access of training data, while regularization methods need training data.
> > > - Achieving lower bias while maintaining or even better prediction performance compared with vanilla methods across many backbones. (shown in Table 1)
> > > - Achieving better fairness-accuracy tradeoff than the regularization method, adversarial debiasing, and Fair Mixup (shown in Figures 2 and 3).
> > >
> > > We believe the above claims are **well-supported** by our experiments. In our paper, we conduct experiments for multiple commonly used fair methods, including adding regularization, adversarial debiasing, and Fair Mixup, with many GNNs and MLP. We also report the fairness-accuracy tradeoff performance across different hyperparameters (shown in Figures 2 and 3). **The results show that FMP trained by cross-entropy loss can achieve better tradeoff performance than these three fair methods across different backbones**. Therefore, our results can support the effectiveness of FMP in terms of tradeoff performance. In other words, bias mitigation can be done during forward propagation with cross-entropy loss, which is highly different from existing fair methods relying on either data preprocessing or fair loss (backpropagation).

---

> > > ### Author Response · Authors · 2022-12-07
> > > **Response to Reviewer Sogu [Part 2/2: Transparency clarifications and quantitative results]**
> > >
> > > Q2: The current version of the paper still fails to demonstrate what is interpretable and/or transparent about FMP, both in terms of paper clarity and qualitative analysis.
> > >
> > > A2: Thank you for your comments. We first find that, for many existing fair methods to obtain the influence of sensitive attributes, the well-trained vanilla model is needed to train from scratch so that we can obtain the difference between the well-trained fair model and the well-trained vanilla model. In other words, the training data is required to obtain the well-trained vanilla model. For our proposed FMP, the influence can be obtained only given a well-trained fair model and test data (the training data is not required). Therefore, We can know how the sensitive attribute influences the mode prediction during the inference stage only with FMP model itself (i.e., FMP can achieve transparency). As for quantitative results, the transparency is binary and can determine based on the definition. We provide the result of sensitive attribute influence in Figure 4.
> > >
> > > We would like to provide more justifications as follows:
> > >
> > > * ***What is the influence of sensitive attributes in the inference stage?***  **The influence of sensitive attributes can be regarded as the difference between the well-trained fair and vanilla model prediction.** It is well-known that most fair methods rely on sensitive attributes to achieve fair prediction. In other words, the sensitive attribute has a critical influence to achieve fair prediction and the prediction is highly different for the vanilla model (trained with vanilla loss and no data preprocessing) and the fair model (trained with fair methods).
> > > * ***What is transparency?*** **Transparency represents that the influence of sensitive attributes can be obtained via the well-trained fair model and test data (without access to the training data).** Based on the definition of the influence of sensitive attribute, it is seen that the fair model and vanilla model are both required to obtain the influence of sensitive attribute. In other words, it is almost intractable to obtain such influence if only having access to the fair model since the training data can not be accessed. Therefore, the vanilla model is **not transparent**. The required resources for estimating the influence should be as less as possible (Less is more). In our paper, the designed FMP can estimate the influence of sensitive attributes during forward propagation without accessing training data. We also revised the statement on transparency in the paper as follows: "Transparency in fairness: Onlookers can obtain the influence of sensitive attribute in the inference stage with only the released well-trained fair model and test data samples".
> > > * ***How does FMP achieve transparency?*** **FMP explicitly and separately uses sensitive attributes in (the last stage) forward propagation, which makes it easy to identify the influence.** In our work, we first use the MLP model to conduct feature transformation (sensitive attribute is not included, we only use node feature matrix $\mathbf{X}$) and use aggregation with skip connection, then debiasing using the sensitive attribute to achieve fair aggregation as the final logit. In other words, the feature transformation, aggregation, and debiasing are separated in FMP, where **the influence of sensitive attribute only happens in the debiasing stage**. Therefore, the influence of sensitive attributes can be directly obtained by probing the input and output of the debiasing stage. The key difference to achieve transparency for FMP is due to the mechanism of achieving fairness. Most existing fair methods rely on either data preprocessing or backpropagation to modify the data or model parameters, which inevitably encode the sensitive attribute information in debiased data or model parameters.

---

> > > ### Comment · Reviewer_Sogu · 2022-12-07
> > > **Response to Authors**
> > >
> > > Dear Authors,
> > >
> > > Thank you for providing answers to my questions. However, I still have some comments:
> > >
> > > * I am still concerned why FMP produces ~10x worse fairness on NBA compared to the FairGNN paper. I acknowledge the setups are different but don't know if this is the only reason or there are some implementation differences, etc.
> > > * I appreciate that the authors elaborated what "influence" is in this paper. I finally understood what they mean by that. In the original submission this was mentioned in a footnote in the introduction. I don't know if it is possible (space-wise, etc.), but it would be good if the authors include some formal definition, too (including in the Method section).
> > > * However, there are still some discrepancies in the definitions. In the rebuttal they say "The influence of sensitive attributes can be regarded as the difference between the well-trained fair and vanilla model prediction". What I understand from the paper, the influence is the difference in the biased/unbiased representations. To me, prediction and representation are not the same.
> > > * Appendix H (and therefore Figure 4) are not referenced from the main text, which I believe might be useful, if the authors think App. H.3 helps with presenting the paper's contributions.
> > > * It is not clear what Figure 4 represents (x/y axis are missing). The figure caption talks about "node representations", the main text talks about "logit layer representation". To me, they are not necessarily the same, e.g., a node representation could be the representation from which the logits are computed.
> > >
> > > Minor:
> > > * In their rebuttal, the authors comment that they "first use the MLP model to conduct feature transformation (sensitive attribute is not included, we only use node feature matrix X)". It is worth clarifying in the paper that the sensitive attribute is not included in the node features. However, even this is not a trivial step because fairness through unawareness does not work (Fairness in machine learning tutorial @ NeurIPS 2017) and there might be some correlations between the features and the sensitive attribute.
> > >
> > > With respect to the authors' efforts and the clarifications they made during the rebuttal phase, I am increasing my score. My decision is final because the changes that were made are non-trivial from (my) reader/reviewer's perspective and I think there is still substantial room for improvement of the paper, at least in terms of clarity (see also my initial comments and questions Q6-7 regarding the Methods section).

---

> > > > ### Author Response · Authors · 2022-12-11
> > > > **Response to Reviewer Sogu**
> > > >
> > > > Thanks to the reviewer for the constructive comments to improve the quality of this work and increase the score. We have revised our paper based on your comments. Please see more detail in the paper https://anonymous.4open.science/r/FMP-AD84/ICLR23_FMP.pdf. Here are some justifications:
> > > >
> > > > * **Difference with FairGNN paper**: Firstly, We consider the scenario that all sensitive attributes are available while FairGNN paper considers the limited sensitive attribute. In FairGNN paper, the sensitive attribute estimator is introduced to tackle limited sensitive attribute cases. In our paper, the sensitive attribute module is not considered. The train/val/test split strategy is also different. We use 0.8/0.1/0.1 for all methods, while FairGNN adopts 0.5/0.25/0.25 in the official code https://github.com/EnyanDai/FairGNN. The key reason leading to high difference in Table 1 is that we report accuracy and fairness metric without fairness method in NBA dataset. In Figure 2, it is seen that the fairness metric of NBA dataset can be low with low accuracy.
> > > > * **Add definitions on "influence"**: We have added more formal description in Section 3.3. We would like to highlight that it is not necessary to narrow down the specific objective for the "influence" of sensitive attributes. It could be (any layer) node representation or prediction.
> > > > * **Add reference for Appendix H.3 and details description in Figure 4**: We have added reference for Appendix H.3 and more description on the x/y axis. We visualize logit layer representation with 2 dimensions since we only consider binary classification tasks in this paper. We revised our paper to make it more clear.
> > > > * **Highlight that the sensitive attribute is not included in the node features**. We have added one footnote in Section 2.1 to describe that the sensitive attribute $\mathbf{s}$ is not included in the node features matrix $\mathbf{X}_{ori}$.
> > > > * **Original questions Q6-7**: We add one more sentence in the first paragraph of Introduction section to specify that we aim to achieve fair prediction. As for the method part, we revised the paper to provide more context on the unified optimization framework for GNNs. Please see Section 2.2 for more details.

---

### Decision · Program_Chairs · 2023-01-20

**Decision:**

Reject

**Justification For Why Not Higher Score:**

Lack of baseline experimental comparisons.

**Justification For Why Not Lower Score:**

N/A

**Metareview: Summary, Strengths And Weaknesses:**

The paper proposes a method for fair message passing in a transparent way via explicitly using sensitive attributes in the forward propagation.
Reviewers appreciated the novelty of the method and that it tackles an under-explored topic, i.e. Fairness in GNNs
However, reviewers pointed out the experiments as a part that can be improved substantially; with lack of comparisons to standard baselines which would be required to convincingly validate the method. This was the key reason for rejection.